# Data-driven modeling predicts gene regulatory network dynamics during the differentiation of multipotential hematopoietic progenitors

**Joanna E. Handzlik**[ID], **Manu**[ID]*

Department of Biology, University of North Dakota, Grand Forks, North Dakota, United States of America

* manu.manu@und.edu

**Data Availability Statement:** There are no primary data in the paper. Gene circuit parameter values are presented in S1 Table. The code for fitting and simulating gene circuits and associated input files

## Abstract

Cellular differentiation during hematopoiesis is guided by gene regulatory networks (GRNs) comprising transcription factors (TFs) and the effectors of cytokine signaling. Based largely on analyses conducted at steady state, these GRNs are thought to be organized as a hierarchy of bistable switches, with antagonism between Gata1 and PU.1 driving red- and white-blood cell differentiation. Here, we utilize transient gene expression patterns to infer the genetic architecture—the type and strength of regulatory interconnections—and dynamics of a twelve-gene GRN including key TFs and cytokine receptors. We trained gene circuits, dynamical models that learn genetic architecture, on high temporal-resolution gene-expression data from the differentiation of an inducible cell line into erythrocytes and neutrophils. The model is able to predict the consequences of gene knockout, knockdown, and overexpression experiments and the inferred interconnections are largely consistent with prior empirical evidence. The inferred genetic architecture is densely interconnected rather than hierarchical, featuring extensive cross-antagonism between genes from alternative lineages and positive feedback from cytokine receptors. The analysis of the dynamics of gene regulation in the model reveals that PU.1 is one of the last genes to be upregulated in neutrophil conditions and that the upregulation of PU.1 and other neutrophil genes is driven by Cebpa and Gfi1 instead. This model inference is confirmed in an independent single-cell RNA-Seq dataset from mouse bone marrow in which Cebpa and Gfi1 expression precedes the neutrophil-specific upregulation of PU.1 during differentiation. These results demonstrate that full PU.1 upregulation during neutrophil development involves regulatory influences extrinsic to the Gata1-PU.1 bistable switch. Furthermore, although there is extensive cross-antagonism between erythroid and neutrophil genes, it does not have a hierarchical structure. More generally, we show that the combination of high-resolution time series data and data-driven dynamical modeling can uncover the dynamics and causality of developmental events that might otherwise be obscured.

are available on GitHub at https://github.com/mlekkha/EryNeu.

**Funding:** This work was supported by award #1615916 from the National Science Foundation (https://www.nsf.gov) to M. The funders had no role in study design, data collection and analysis, decision to publish, or preparation of the manuscript.

**Competing interests:** The authors have declared that no competing interests exist.

## Author summary

The supply of blood cells is replenished by the maturation of hematopoietic progenitor cells into different cell types. Which cell type a progenitor cell develops into is determined by a complex network of genes whose protein products directly or indirectly regulate each others' expression and that of downstream genes characteristic of the cell type. We inferred the nature and causality of the regulatory connections in a 12-gene network known to affect the decision between erythrocyte and neutrophil cell fates using a predictive machine-learning approach. Our analysis showed that the overall architecture of the network is densely interconnected and not hierarchical. Furthermore, the model inferred that PU.1, considered a master regulator of all white-blood cell lineages, is upregulated during neutrophil development by two other proteins, Cebpa and Gfi1. We validated this prediction by showing that Cebpa and Gfi1 expression precedes that of PU.1 in single-cell gene expression data from mouse bone marrow. These results revise the architecture of the gene network and the causality of regulatory events guiding hematopoiesis. The results also show that combining machine learning approaches with time course data can help resolve causality during development.

## 1 Introduction

Cell-fate decisions during hematopoiesis are thought to be made by transcriptional gene regulatory networks (GRNs) [1–3], which are comprised of genes that influence each others' expression through their products. The genetic architecture, by which we mean the regulators of genes, whether each regulator activates or represses, as well as the quantitative strength of regulation, of hematopoietic GRNs is not fully understood. Hematopoietic cell-fate choice has often been interpreted in the context of a simple network motif, the bistable switch [3–5]. In the bistable switch model, two TFs repress each others' expression and cell-fate is chosen in a cell-autonomous manner when small stochastic fluctuations cause the system to shift to one of two steady states corresponding to the alternative cell fates. For example, the choice between the red- and white-blood cell fates is thought to be made by mutual repression between two transcription factors (TFs), Gata1 and PU.1 (encoded by *Spi1*) [4]. Similar bistable switches have been proposed for other binary cell-fate choices in hematopoiesis [2] and more generally in development [6].

  A number of recent developments suggest that the bistable switch framework might be insufficient to explain cell-fate choice and that hematopoietic GRNs have a densely interconnected architecture. Network reconstructions based on genome-wide gene expression data reveal large modules of co-regulated genes [7] and genome-wide TF binding data show that most regulatory regions are co-bound by multiple TFs [8, 9]. A second issue is that the bistable-switch hypothesis is anchored in a developmental sequence of discrete binary cell-fate decisions with well-defined intermediate progenitors. Single-cell RNA sequencing data imply however that cellular states during hematopoiesis are situated along a continuum and may not involve binary decisions [10, 11]. Bistable switches, such as Gata1-PU.1, were inferred from genetic and biochemical analyses conducted at steady state, which lack information about the dynamics and causality of events. For instance, tracking the expression dynamics of fluorescently tagged Gata1 and PU.1 in live cells suggests that rather than initiating lineage choice, the divergent expression of the two proteins is itself a consequence of as-yet-unknown upstream regulatory events [12]. Finally, the cell-autonomous bistable-switch framework

cannot integrate and account for the instructive influence of cytokines on hematopoietic differentiation [13, 14].

Here we take an alternative approach to inferring the genetic architecture and dynamics of the red- and white-blood cell-fate decision. Our approach utilizes a data-driven predictive modeling methodology called gene circuits [15, 16]. Gene circuits determine the time evolution of protein or mRNA concentrations using coupled nonlinear ODEs in which synthesis is represented as a switch-like function of regulator concentrations. The data can be derived from a wide variety of experiments, ranging from genome-wide studies of unperturbed development to narrower studies involving targeted perturbations. The values of the free parameters define the regulatory influences among the genes in the network. Gene circuits do not presuppose any particular scheme of regulatory interactions, but instead determine it by estimating the values of the parameters from quantitative data using global nonlinear optimization techniques [17–20]. Gene circuits infer not only the topology of the GRN but also the type, either activation or repression, and strength of interactions. Most importantly, the inference procedure yields ODE models that can be used to interrogate the dynamics and causality of regulatory events during differentiation as well as to simulate and predict the consequences of developmental perturbations [21–24].

We inferred the genetic architecture and gene regulation dynamics underlying red- and white-blood cell differentiation using gene circuit models comprising 12 genes. The gene circuits included receptors and effectors of cytokine siganling in addition to well-known lineage specifying TFs, such as Gata1 and PU.1, so that they could incorporate the potential influence of cytokines. The gene circuits were trained on publicly available high temporal resolution genome-wide gene expression data acquired during the differentiation of an inducible cell line, FDCP-mix [4, 25], into erythrocytes and neutrophils. Most of the inferred pair-wise regulatory interactions were consistent with available empirical evidence. The models also correctly predicted the effect of knock-out, knock-down, and overexpression of key TFs both qualitatively and quantitatively. The genetic architecture of the models, instead of being hierarchical, is densely interconnected and features extensive cross-repression between genes expressed in different lineages. Furthermore, analysis of the model showed that *Spi1* upregulation occurred in the latter half of neutrophil differentiation, which was driven instead by two other TFs expressed in the neutrophil lineage, C/EBP$\alpha$ and Gfi1. We tested this prediction of the model by inspecting the sequence of gene upregulation during neutrophil differentiation in a single-cell RNA-seq dataset [11] from mouse bone marrow. These data confirmed that *Cebpa* and *Gfi1* upregulation precede that of *Spi1 in vivo*.

## 2 Results

### 2.1 Data-driven modeling of gene expression dynamics during the differentiation of FDCP-mix cells

**2.1.1 Gene circuit models.** A gene circuit [22] computes the time evolution of mRNA concentrations of a network of interacting genes by solving the coupled ordinary differential equations (ODEs)

$$\frac{dx_i^l}{dt} = R_i S\left(\sum_{j=1}^{N} T_{ij} x_j^l + b_i c^l + h_i\right) - \lambda_i x_i^l, \tag{1}$$

where $x_i^l(t)$ is the concentration of the mRNA of gene $i$ at time $t$ in lineage (or condition) $l$, and $N$ is the total number of genes in the model. The synthesis rate depends on the concentrations of a gene's regulators through sigmoidal regulation-expression function

$S(u) = \frac{1}{2}\left(u/\sqrt{(u^2+1)} + 1\right)$. $S(u)$ determines the fraction of the maximum synthesis rate $R_i$ attained by the gene given the total regulatory input $u = \sum_{j=1}^{N} T_{ij}x_j^l + b_ic^l + h_i$. The first term of $u$, $\sum_{j=1}^{N} T_{ij}x_j^l$, represents the regulation of gene $i$ by the other genes in the network. Positive and negative values of $T_{ij}$ signify activation and repression of gene $i$ by gene $j$ respectively. The regulation of gene $i$ by factors specific to the condition $l$ that have not been explicitly represented in the model is described by the second term of $u$, $b_ic^l$, where $c^l$ is −1, 0, or 1 for neutrophil, progenitor, and erythroid conditions respectively. The threshold $h_i$ determines the basal synthesis rate and $\lambda_i$ is the degradation rate of mRNA for gene $i$. Training gene circuit models on quantitative gene expression data results in estimates of the values of these parameters. Estimates of the genetic inteconnectivity coefficients ($T_{ij}$) allows the inference of the genetic architecture of the GRN.

The sigmoid regulation-expresssion function allows the synthesis rate of a target to change with a regulator's concentration in either a gradual or a sharp manner, depending on the magnitude of the genetic interconnectivity $T_{ij}$. If the magnitude is small, then the synthesis rate will change gradually as the regulator's concentrations is varied. If the magnitude is large, small changes in regulator concentration can lead to sharp changes in synthesis rate. In the latter scenario, sharp changes occur when the total regulatory input crosses zero and hence the regulator does not have a fixed threshold concentration, which can vary depending on the contributions of other regulators.

**2.1.2 Specification of a gene circuit model for erythrocyte-neutrophil differentiation.**
We constructed a gene circuit model comprising 12 main lineage-specifying TFs and cytokine receptors implicated in erythrocyte-neutrophil differentiation. *Tal1* and *Gata2* are expressed in pluripotent stem cells and are necessary for the differentiation of multiple lineages including erythrocytes [26–31]. *Gata1*, its partner *Zfpm1*, which encodes the Fog1 protein, and *Klf1* are necessary for erythroid and megakaryocytic differentiation [2, 32–37]. All white blood-cell lineages are absent in the bone marrow of *Spi1*$^{-/-}$ knockout mice [38], the products encoded by *Cebpa* and *Gfi1* specify the neutrophil cell fate [3, 39], and the TF encoded by *Stat3* acts downstream of GCSF signaling [40]. Previous work has suggested that the expression level of cytokine receptors can influence the activation of lineage specifying TFs [41]. We included three genes, *Epor*, *Csf3r*, and *Il3ra*, encoding the cytokine receptors Epor, GCSF-R, and the alpha subunit of the IL3 receptor respectively [42] in order to detect such potential regulatory mechanisms. Although all of these genes are well-known participants in erythrocyte-neutrophil differentiation, the precise genetic architecture of the network remains to be determined.

While there are other genes known to be important for the specification of these cell fates, we limited the number of genes to 12 in order to minimize the risk of overfitting and to complete model fitting in a reasonable amount of time. Increasing the number of genes increases the number of free parameters in the model and these extra degrees of freedom increase the chances that the model will be overfit, resulting in poor predictive ability. With the training data used here, the 12 gene model has a three-fold excess of datapoints over free parameters, which makes overfitting unlikely.

**2.1.3 Time-series data for training the gene circuits.**   We trained the gene circuit on May *et al.*'s high temporal-resolution dataset [25] of genome-wide gene expression during erythrocyte-neutrophil differentiation. May *et al.* [25] utilized FDCP-mix cells [34] which are maintained in a multipotent state in the presence of IL3 and can be induced to differentiate into erythrocytes or neutrophils by culturing in low IL3, Epo, and hemin or GCSF and SCF respectively. In the rest of the paper, we refer to the culture of FDCP-mix cells in low IL3, Epo, and hemin as erythrocyte conditions and culturing in GCSF and SCF as neutrophil conditions. The dataset comprises genome-wide gene expression measurements at 30 time points during

the 7-day course of differentiation towards either cell fate, with sampling frequency reducing from once every two hours during the first day to once in three days during the last three days.

The trajectories of gene expression for the modeled genes (Fig 1) exhibit rich temporal dynamics. Whereas the expression of some genes, such as *Klf1* and *Gfi1*, diverges between erythroid and neutrophil conditions during the first few hours of differentiation, the expression of genes such as *Il3ra*, *Gata2*, and *Spi1* does not diverge until 2–3 days in to the differentiation. Besides timing, the genes differ also in the magnitude of change during the course of differentiation. Although all the genes change expression significantly over 7 days, *Il3ra* is upregulated $\sim$2-fold in the neutrophil condition while *Csf3r* is upregulated $\sim$230-fold in the neutrophil condition. With the exception of *Gata2*, the expression patterns of the remaining genes are consistent with those in murine bone marrow at a qualitative level (S1 Fig). While *Gata2* is upregulated in both conditions in FDCP-mix cells, it is downregulated along both the erythrocyte and neutrophil lineages in data from bone marrow. Lastly, all genes except *Gata2* demonstrate an "either-or" pattern of regulation in FDCP-mix cells, being upregulated in one condition, while being downregulated in the other (Fig 1).

**2.1.4 Training the gene circuits on time-series gene expression data.** We trained the gene circuits on May et al.'s time series data [25], with initial conditions specified by gene expression in progenitor cells, using a global nonlinear optimization method called Parallel Lam Simulated Annealing (PLSA; [17, 22]). PLSA is a stochastic method and results in a distinct set of parameters each time a gene circuit is inferred from data. In order to ensure that our analysis was not influenced by any idiosyncrasy of a particular model, we inferred 100 independent gene circuit models, and chose 71 that met our goodness-of-fit of criteria (Section 4) for further analysis.

**2.1.5 Simulation of the GRN during FDCP-mix erythrocyte-neutrophil differentiation.** The output of the 71 analyzed gene circuits agreed with data to within experimental error for all 12 genes and the vast majority of time points (Fig 1). The sole exception was that the models did not reproduce a spike in *Cebpa* expression occurring around the 70 hour time point, although it is unclear whether this spike is genuine or the result of experimental error. Models trained on randomly shuffled data fit the data poorly (Section 4), implying that the fits to the empirical data are statistically significant (S2 Fig). We also checked how sensitive the model is with respect to perturbations in initial conditions and found that model output was robust to perturbations of up to ±70% (S3 Fig). Consistent with the general agreement with the data, the models' outputs reproduce all the essential dynamical features of the data—the either-or differential expression, gene-specific timing of expression divergence, and gene-to-gene variation in the dynamic range of expression.

## 2.2 Gene circuits predict the consequences of genetic perturbations

Having obtained gene circuits that are able to quantitatively reproduce the observed time series data, we next tested whether the models could predict the outcomes of experimental treatments *de novo*, that is, without being trained on the data from the experiments. We simulated two kinds of experiments using the gene circuits. The first class are knockouts of *Gata1* and *Spi1*, experiments that were not carried out in FDCP-mix cells but in mice or other cell types. One should not expect the model to predict the outcomes of such knockout experiments at a quantitative level since the model was neither trained on the data from these cell types nor were all of its state variables measured in the experiments. Therefore, we compare model predictions with the results of knockout experiments at a qualitative level. The second class of experiments involved the knockdown or overexpression of key gene products followed by genome-wide expression profiling conducted by May *et al.* in FDCP-mix cells [25]. Simulation

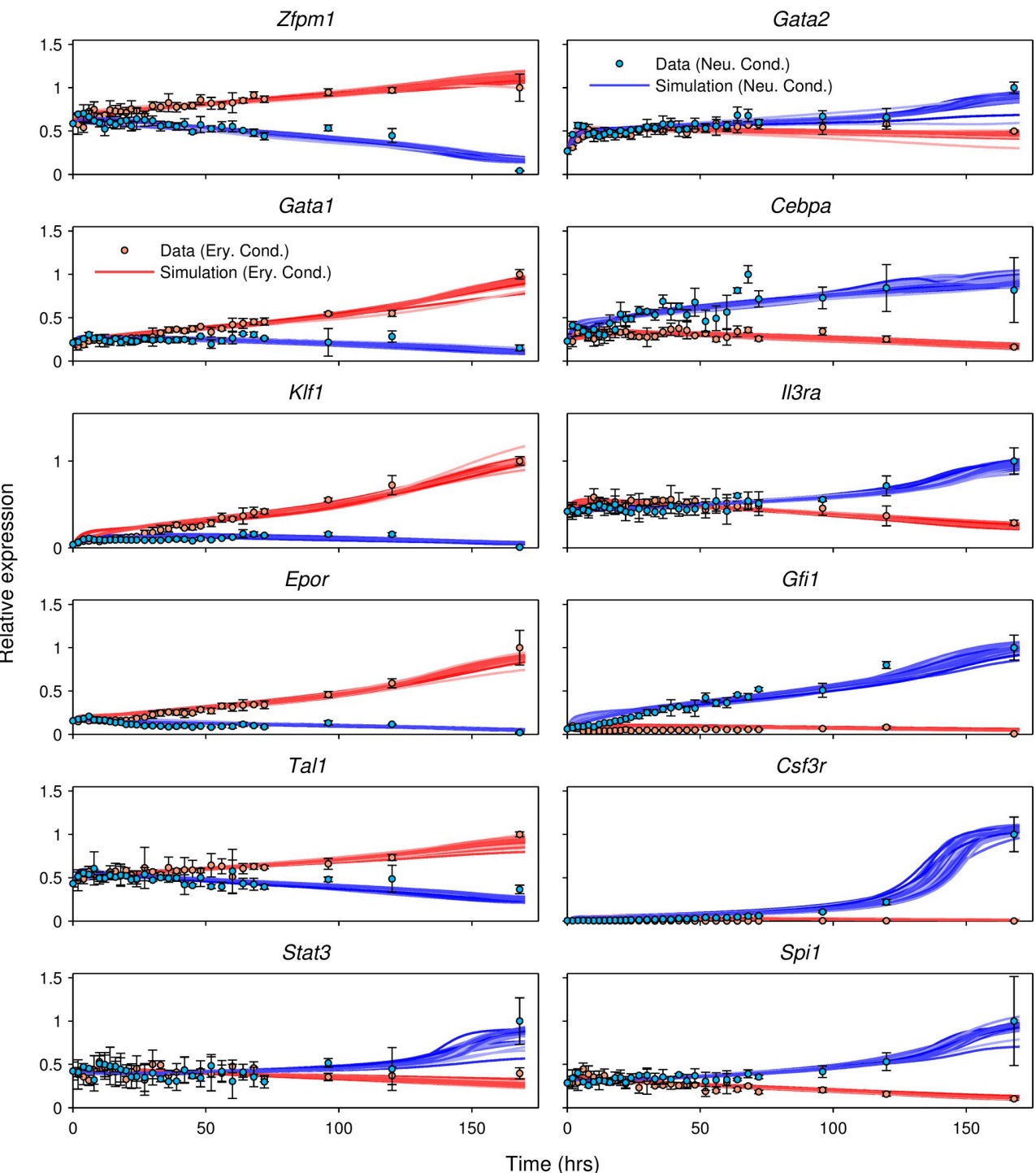

**Fig 1. Gene expression time series data vs. model output.** Mean microarray gene expression measurements and model output for the 12 modeled genes are plotted as circles and lines respectively. Here, and in the following figures, relative expression of a gene is given by the ratio of its expression to its maximum expression across all conditions and time points. Errors bars show standard deviation over 3 replicates. The output of the 71 models that met the goodness-of-fit criteria (Section 4) are shown simultaneously. Data and model output for FDCP-mix cells cultured in low IL3, Epo, and hemin, referred to as the erythrocyte condition hereafter, are shown in red. Data and model output for FDCP-mix cultured in GCSF and SCF, referred to as the neutrophil condition hereafter, are shown in blue.

of these perturbations may be compared to experiments at a quantitative level since they share the experimental system and all of the model's state variables were measured.

**2.2.1 Simulation of *Spi1* and *Gata1* knockout.**   We simulated *Spi1* knockout by setting its initial expression and maximum synthesis rate to zero (Section 4). The consequences of this perturbation differed by condition (Fig 2). In erythrocyte conditions, although the change was more rapid in the mutant, the expression of all genes moved in the same direction and attained very similar values on day 7 as the wildtype. The model predicted therefore that erythrocyte differentiation is largely unperturbed in *Spi1* mutants, which matches experimental observations from *Spi1* knockout mice [38]. Gene expression temporal profiles differed markedly between mutant and wildtype in neutrophil conditions however, and changed very little from their initial values. A lack of change in gene expression implies that cells are arrested in a progenitor state in the *Spi1* mutant during neutrophil differentiation. This prediction is supported by the observations that *Spi1* knockout mice lack mature white-blood cells [38] and that their bone marrow contains IL3-dependent granulocyte-monocyte progenitors (GMPs) [43, 44], while disruption of *Spi1* in mouse granulocyte/monocyte-committed progenitors prevents their maturation but not proliferation [45].

The results of *Gata1* knockout (S4 Fig) were opposite to those of *Spi1* knockout. In neutrophil conditions, the expression of all genes changed in the same direction and reached the same endpoints as in wildtype, albeit more rapidly, implying that neutrophil differentiation is not affected by *Gata1* mutation. In erythrocyte conditions however, gene expression of all genes did not change much from initial conditions, implying an arrest in the progenitor state. These predictions match the empirical results that embryonic stem cells (ESCs) lacking *Gata1* undergo developmental arrest at the proerythroblast stage [46] and that *Gata1*-null ESCs cultured in the presence of Epo resemble proerythroblasts [47].

**2.2.2 Simulation of knockdown and overexpression experiments in FDCP-mix cells.** We simulated the knockdown of *Spi1* and *Gata2* in FDCP-mix cells and compared model output to the changes in gene expression observed in experiment [25]. Since the knockdown was performed in self-renewing IL3 conditions, we set the lineage condition parameter to zero (Section 4) and simulated knockdown by reducing the synthesis rate of either gene and computing the solution until equilibrium was achieved. Since the knockdown efficiency achieved in the experiment is unknown, we set the synthesis rate to a value that results in a fold change in the expression of the targeted gene—*Spi1* or *Gata2*—that matches the empirically observed value. Therefore, we "fit" the knockdown model to the expression of the targeted gene to predict the changes in the expression of the remaining eleven genes. Finally, this analysis—and all subsequent analyses—were performed using one representative model (model #66) out of the 71 that matched the goodness-of-fit criteria (Section 4).

There is strong agreement between prediction and observation for *Spi1* knockdown (Fig 3). Consistent with the well known regulatory role of PU.1, the model predicted the upregulation and downregulation of the erythrocyte and neutrophil lineage genes respectively, which matched the pattern of gene expression observed in the experiment. The only exception was *Il3ra*, which was predicted by the model to be slightly downregulated but in fact did not change in expression. In contrast to the results with *Spi1*, the model was unable to predict the consequences of *Gata2* knockdown (Fig 3), suggesting that aspects of *Gata2*'s regulation were inferred poorly by model training. This is corroborated by the fact that many of the Gata2-related regulatory parameters were poorly constrained (Fig 4).

The overexpression experiments were simulated differently than knockdown experiments since the induction of the ERT fusion proteins by OHT does not change their mRNA expression directly but changes their TF activity, instead. Since the genetic interconnectivity matrix elements parameterize the activity of the TFs in gene circuits, we simulated the induction of

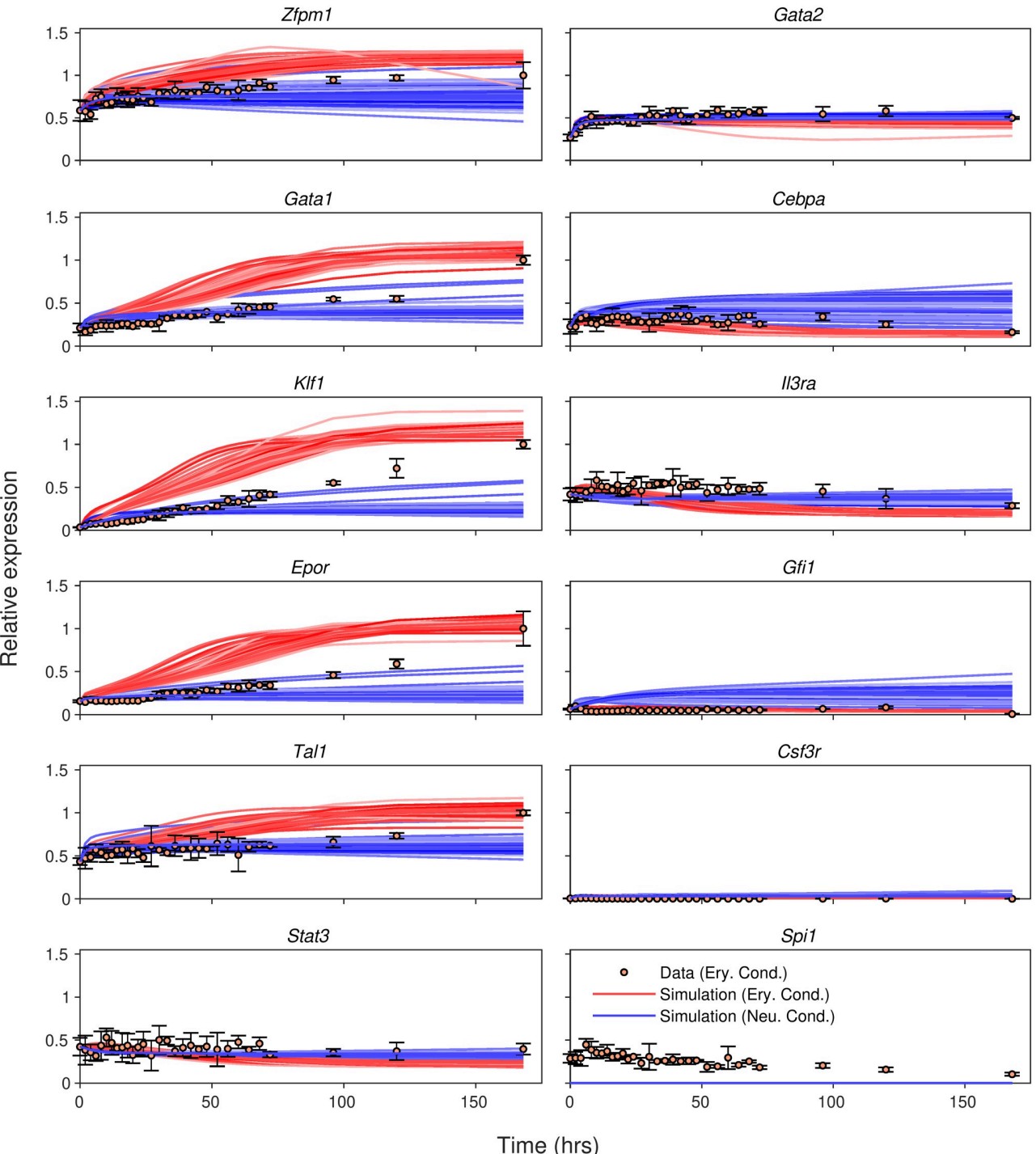

**Fig 2. Simulation of *Spi1* knockout.** *Spi1* knockout was simulated in all 71 models that met the goodness-of-fit criteria. Their output is plotted as lines. The symbols and colors are the same as Fig 1.

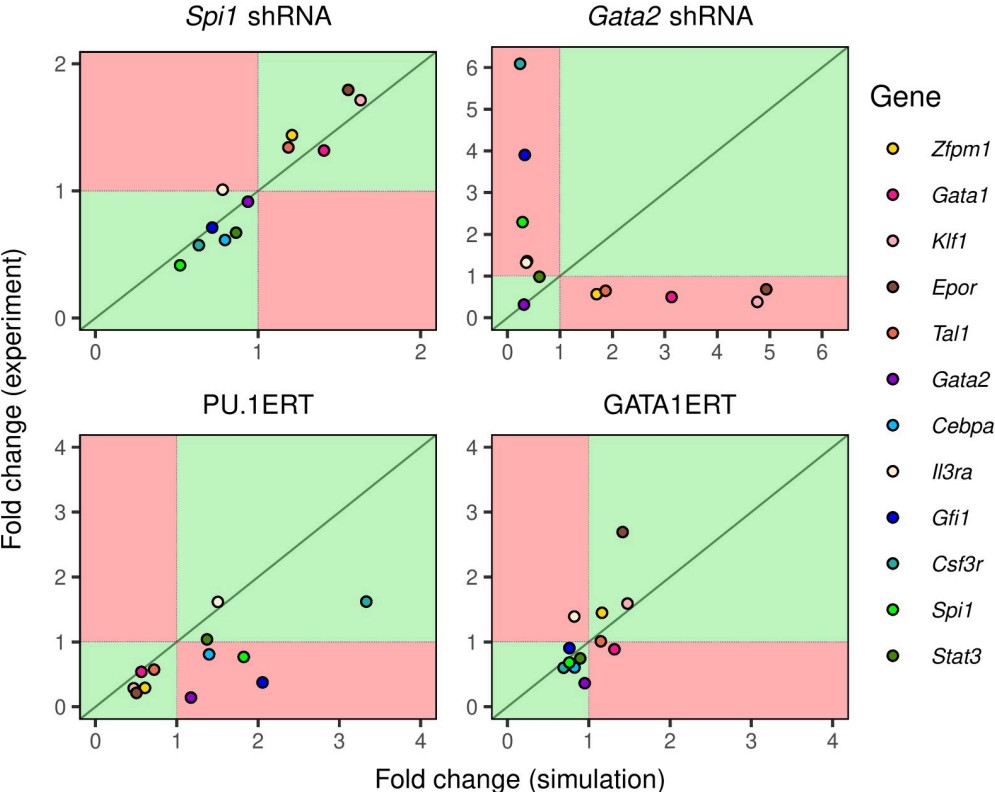

**Fig 3. Simulation of knockdown and overexpression of key transcription factors in FDCP-mix cells.** The fold change in gene expression in simulations of *Spi1* and *Gata2* knockdown (top two panels) or PU.1 and Gata1 overexpression (bottom two panels) is plotted against the fold change observed in experiment. The dotted lines correspond to no change so that points in the green quadrants indicate qualitative agreement and points in the red quadrants indicate qualitative disagreement between prediction and observation. The green line represents perfect quantitative agreement between prediction and observation.

ERT protein activity by OHT by adding a bias term to the total regulatory input of each gene. The bias term of each gene is proportional to the interconnectivity element through which the gene is regulated by the overexpressed gene (Section 4). Similar to the knockdown experiments, the proportionality constant is unknown and was determined by fitting the overexpression model to the expression of one of the genes. Finally, we did not fit to the expression of the overexpressed gene since the observed mRNA includes an unknown contribution from the ERT fusion transgene.

The model was able to correctly predict the change in expression of all the genes except *Il3ra* in the GATA1ERT experiment (Fig 3). The quantitative agreement between model prediction and experiment was also good with the exception of *Epor*, for which a $\sim$ 1.5-fold upregulation was predicted while a $\sim$ 3-fold upregulation was observed. In the PU.1ERT experiment, the model predicted the change in expression of all genes except *Cebpa*, *Gfi1*, and *Gata2*. Whereas the model predicted an upregulation of these genes upon PU.1 overexpression, these genes were found to be downregulated in the actual experiment. The downregulation of *Gfi1* and *Cebpa* observed in experiment is inconsistent with the known role of PU.1 as an activator of these white-blood cell lineages genes [48–51] as well as their downregulation upon *Spi1* knockdown. This inconsistency could be the result of PU.1 overexpression promoting a macrophage gene expression program by repressing neutrophil genes indirectly via *Egr1/*

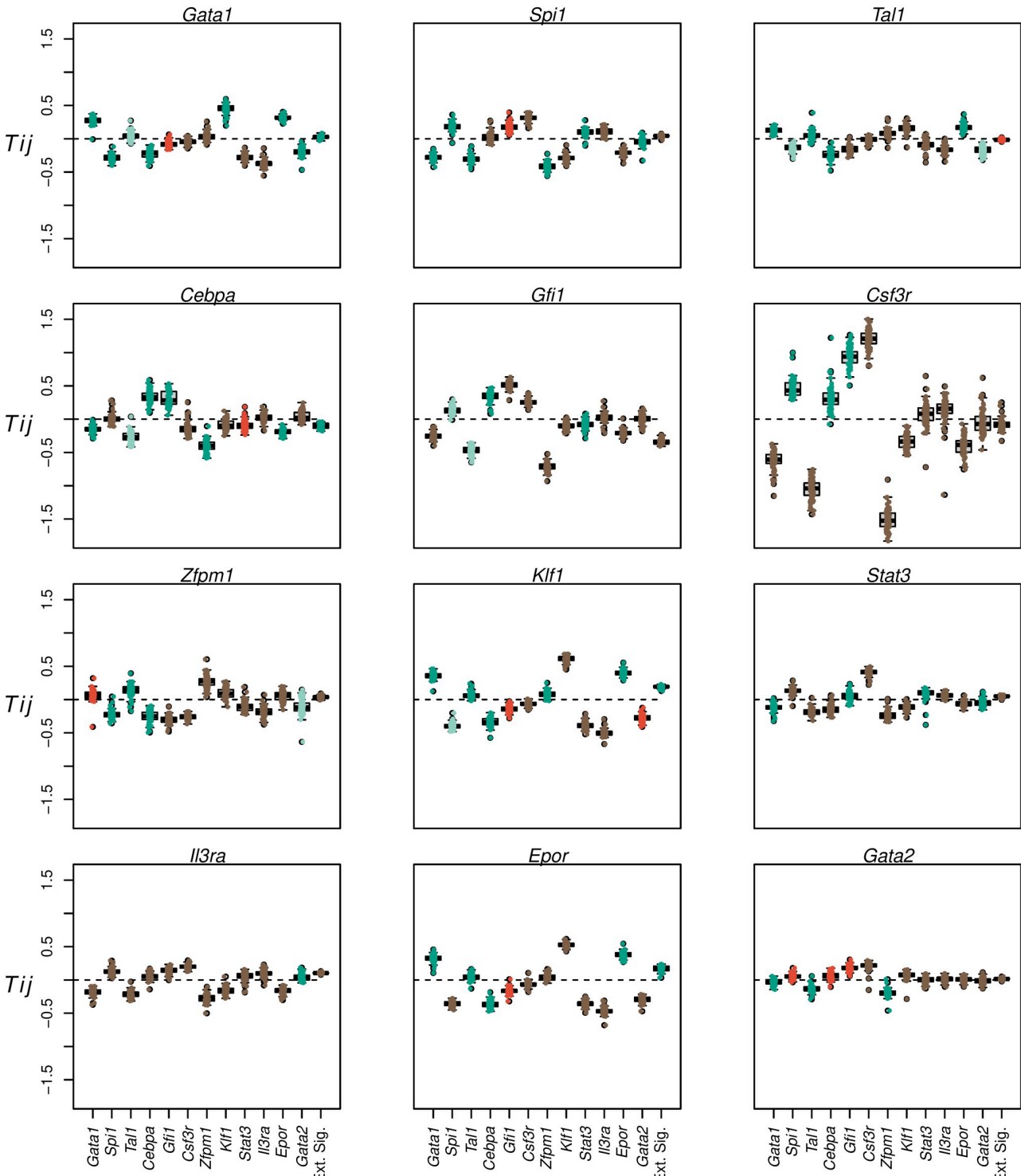

**Fig 4. Inferred genetic architecture.** The distribution of each genetic interconnectivity parameter ($T_{ij}$) over the ensemble of 71 models is shown as a box plot. The distribution of the each regulatory parameter representing the influence of cytokine conditions ($b_i$) is shown as a box plot ("Ext. Sig."). In the box plots, the box lines are the first quartile, median, and third quartile. The whiskers extend to the most extreme values lying within 1.5 times the interquartile range. Individual parameter values inferred by the models are shown as circles overlaid on the box plots. Each panel shows the regulation of a particular target. Positive and negative values of $T_{ij}$ indicate activation and repression respectively. Positive values of $b_i$ indicate activation by Epo and repression by GCSF while negative values indicate activation by GCSF and repression by Epo. Activation is inferred if the first quartile of the distribution is positive, while repression is inferred if the third quartile is negative. The type of regulation is considered to be poorly constrained when

the interquartile range spans negative and positive values. The parameters whose inferred sign agrees with prior empirical evidence (S2 Table) are marked as dark green while those that are contradictory are marked as red. The parameters for which there is empirical evidence for an interaction but the type of interaction, activation or repression, is not known are marked as light green. The parameters for which we were unable to find experimental evidence, the experiments yielded negative results, or the sign was unconstrained are marked as brown.

*2* [3] or *Irf8* [52]. The misprediction of *Cebpa* and *Gfi1* expression in PU.1ERT could therefore be a consequence of omitting macrophage lineage genes in the model.

## 2.3 Erythrocyte-neutrophil GRN architecture is non-hierarchical and evolves in time

Having verified that the inferred models have predictive ability, we next determined the architecture of the GRN implied by the values of the genetic interconnectivity parameters, $T_{ij}$. $T_{ij}$ determines how the product of gene *j* regulates gene *i*, where positive or negative values denote activation or repression respectively. The distributions of the majority of interconnectivity parameters across the ensemble of 71 analyzed models were well constrained and distinguishable as either activation or repression (Fig 4 and S1 Table). For example, the positive values of $T_{Gata1 \rightarrow Gata1}$ (Fig 4A) and $T_{Spi1 \rightarrow Spi1}$ (Fig 4B) in all but one model implies that both genes autoactivate while the negative values of $T_{Gata1 \rightarrow Spi1}$ (Fig 4B) and $T_{Spi1 \rightarrow Gata1}$ (Fig 4A) in all analyzed models implies that the two genes repress each other. We compared the inferred genetic interconnections to published empirical evidence (Fig 4 and S2 Table). The model inferred the correct role, activation or repression, for 58 of the 69 interconnections that we found empirical evidence for. The vast majority of the interconnections have not been previously examined and the model therefore implies novel inferences about the genetic architecture of the network.

The experimental evidence was inconclusive or conflicting in some instances (S2 Table). Notably, the model inferred that Gfi1 activates *Spi1*, upregulated during FDCP-mix neutrophil differentiation, and represses *Gata1*, *Klf1*, and *Epor*, genes downregulated during FDCP-mix neutrophil differentiation (Fig 1). These model inferences are supported by single-cell RT-qPCR data that show that *Gfi1* expression is positively correlated with *Spi1* expression in GMPs, LMPPs, and HSCs, while it is negatively correlated with *Gata1* expression in HSCs and GMPs [53]. Furthermore, Gfi1 is known to cooperate with C/EBPε to activate neutrophil genes [54, 55]. Contradicting the model's inferences and the above evidence, *Spi1* is upregulated in MPPs from *Gfi1*$^{-/-}$ mice [56, 57] while *Gata1*, *Klf1*, and *Epor* are downregulated in bone-marrow cells from *Gfi1*$^{-/-}$ mice [58]. The conflicting evidence and lack of agreement between the model and data may be a result of the pleiotropic roles that *Gfi1* plays in both HSC maintenance and neutrophil development [59]. As noted in the previous section, the regulatory parameters of *Gata2*, another gene acting pleiotropically in HSCs, the erythroid-megakaryocytic lineage, and the myeloid lineage [26, 60], were poorly or incorrectly inferred. These inconsistencies were, however, a small proportion of the total inferences and there is overall good agreement between model inference and empirical evidence (Fig 4 and S2 Table).

The genetic architecture of the network, in fact, changes in time since the strength of the regulation of one gene by the products of another gene depends on the concentration of the latter, which evolves during the differentiation process. In order to gain insight into this "dynamical GRN", we computed the time-dependent regulatory contribution, given by the product of the genetic interconnectivity parameters by the concentrations of the cognate regulators ($T_{ij} \cdot x_j^l(t)$), for all pairs of regulators and targets in the model. The GRN may then be represented as a graph in which each gene is a node and the type—activation or repression—and time-dependent strength of regulation between each gene pair is an edge (Fig 5).

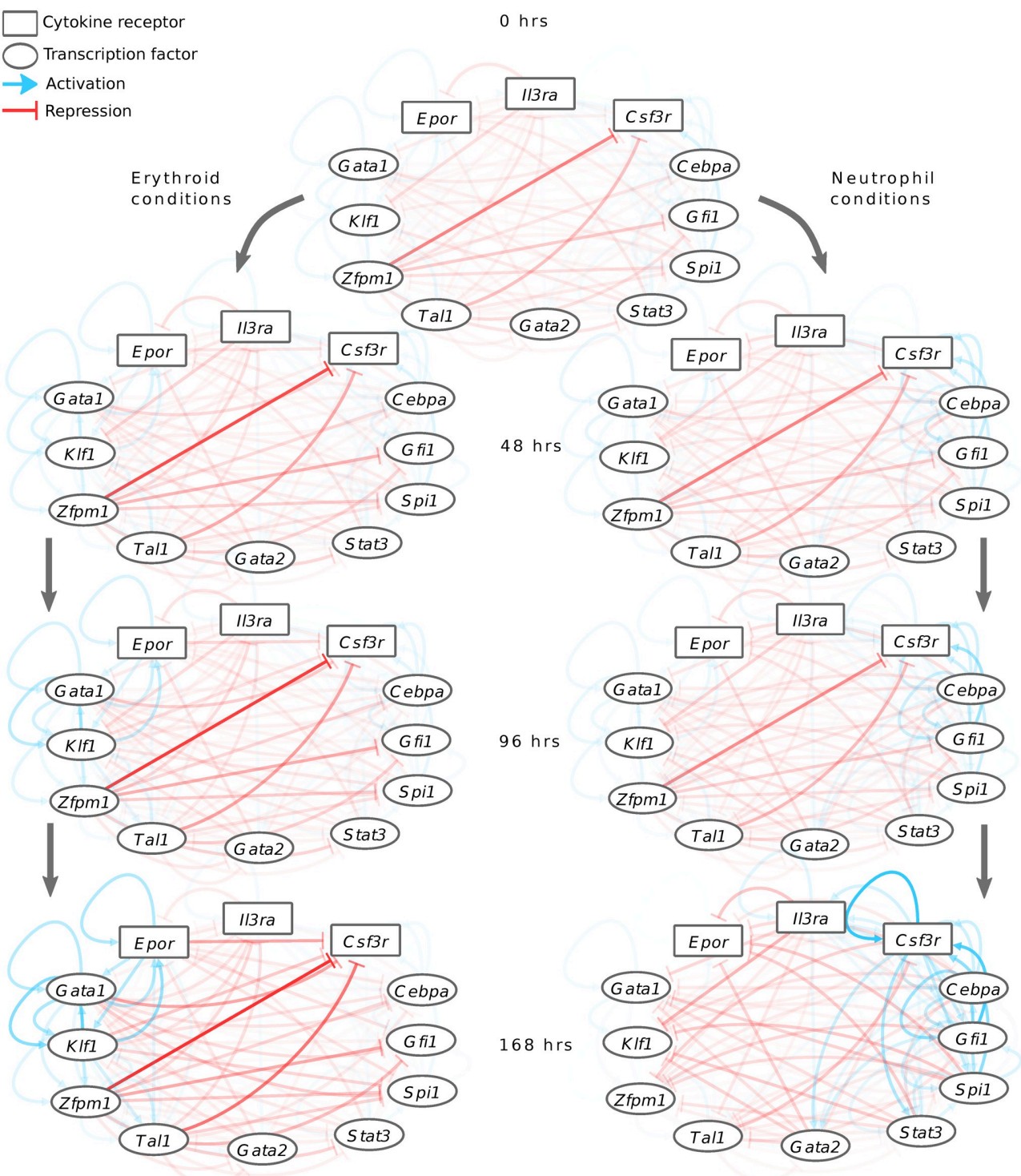

**Fig 5. The time evolution of the inferred GRN.** The GRN is depicted as a graph at different time points during differentiation in both erythrocyte and neutrophil conditions. The contributions of each regulator to the regulation of its targets, given by the product of the pairwise genetic interconnectivity parameter and the regulator's concentration, are shown as edges from the regulator to the targets. Blue and red edges correspond to activation and repression respectively, while the opacity of the lines indicates the strength of regulation. The maximum opacities of activation and repression have been normalized to 1 separately.

The erythrocyte-neutrophil network inferred by the model from FDCP-mix data is densely interconnected with genes associated with the erythrocyte lineage repressing genes of the neutrophil lineage and vice versa. This conclusion is in agreement with other analyses based on genome-wide gene expression data [7] and contrasts the view that the genetic architecture consists of a hierarchy of bistable switches [2]. The time evolution of the network reveals two broad principles. First, there is a preponderance of repressive interactions at earlier time points during the differentiation suggesting that the cell-fate decision is dictated by loss of repression rather than a gain of activation. Conversely, activation between co-expressed genes gains prominence at later time points, suggesting that activation mainly reinforces the decision once it has been made.

## 2.4 Gene circuits predict that C/EBP$\alpha$ and Gfi1 drive neutrophil development in FDCP-mix cells

How each gene in the network is regulated is, as discussed earlier, not static but changes as the concentrations of its regulators evolve in time during differentiation. We reasoned that the temporal dynamics of gene regulation could provide insight into the causality of the regulatory events underlying differentiation. The temporal dynamics of gene regulation can be analyzed by "looking under the hood" of the gene circuit model and decomposing the total regulatory input for each gene into contributions from individual regulators (Fig 6; see Section 4 for details). In Fig 6, the total regulatory input (dotted black line) is plotted in time. A gene is at half its maximum activation when the total regulatory input is zero and thus the time at which this happens (black vertical lines) serves as a marker to order the sequence in which genes turn on or off as differentiation proceeds. The contributions of repressors and activators are shown as shaded sections above and below the total regulatory input respectively. The regulators accounting for the up- or down-regulation of a gene can be determined by noting their contribution to the change in the total regulatory input. For example, the bulk of the change in *Cebpa*'s regulatory input from the start of neutrophil differentiation to reaching half-max expression is the result of autoactivation (light blue) and activation by Gfi1 (dark blue; Fig 6).

Several observations can be made regarding the temporal dynamics of gene regulation during erythrocyte-neutrophil differentiation (Fig 6). All the genes are in a partially repressed state, since the negative contribution from repressors is greater than the positive contribution from activators, in undifferentiated FDCP-mix cells. This is reminiscent of multilineage transcriptional priming [3, 61, 62]—the low-level expression of genes from multiple lineages in multipotential progenitors. What accounts for the repression varies by the target gene. Genes downregulated in neutrophil conditions, *Gata1*, *Zfpm1*, *Klf1*, *Tal1*, and *Epor*, are repressed by several genes of small effect. Genes downregulated in erythrocyte conditions however, *Spi1*, *Cebpa*, *Gfi1*, *Stat3*, *Gata2*, *Il3ra*, and *Csf3r*, are mainly repressed by a combination of *Zfpm1* and *Tal1*.

During erythrocyte differentiation, all the upregulated genes are activated more or less simultaneously since they reach half-max activation in a short $\sim 30$ hour window (Fig 6). Upregulation of the genes involves both the loss of repression as well as increased activation (Fig 6). The three main activating influences are *Gata1*, *Klf1*, and *Epor*. The first two are well known activators of erythrocyte genes, while the activating influence of *Epor* implies that upregulation of the receptor's gene expression provides positive feedback, indirectly, to the TFs driving erythroid differentiation.

In contrast to erythrocyte differentiation, the sequence of activation of genes during neutrophil differentiation is spread out over $\sim 100$ hours (Fig 6). Surprisingly, *Spi1* is one of the last genes in the activation sequence, reaching half-max activation around day 5 of the

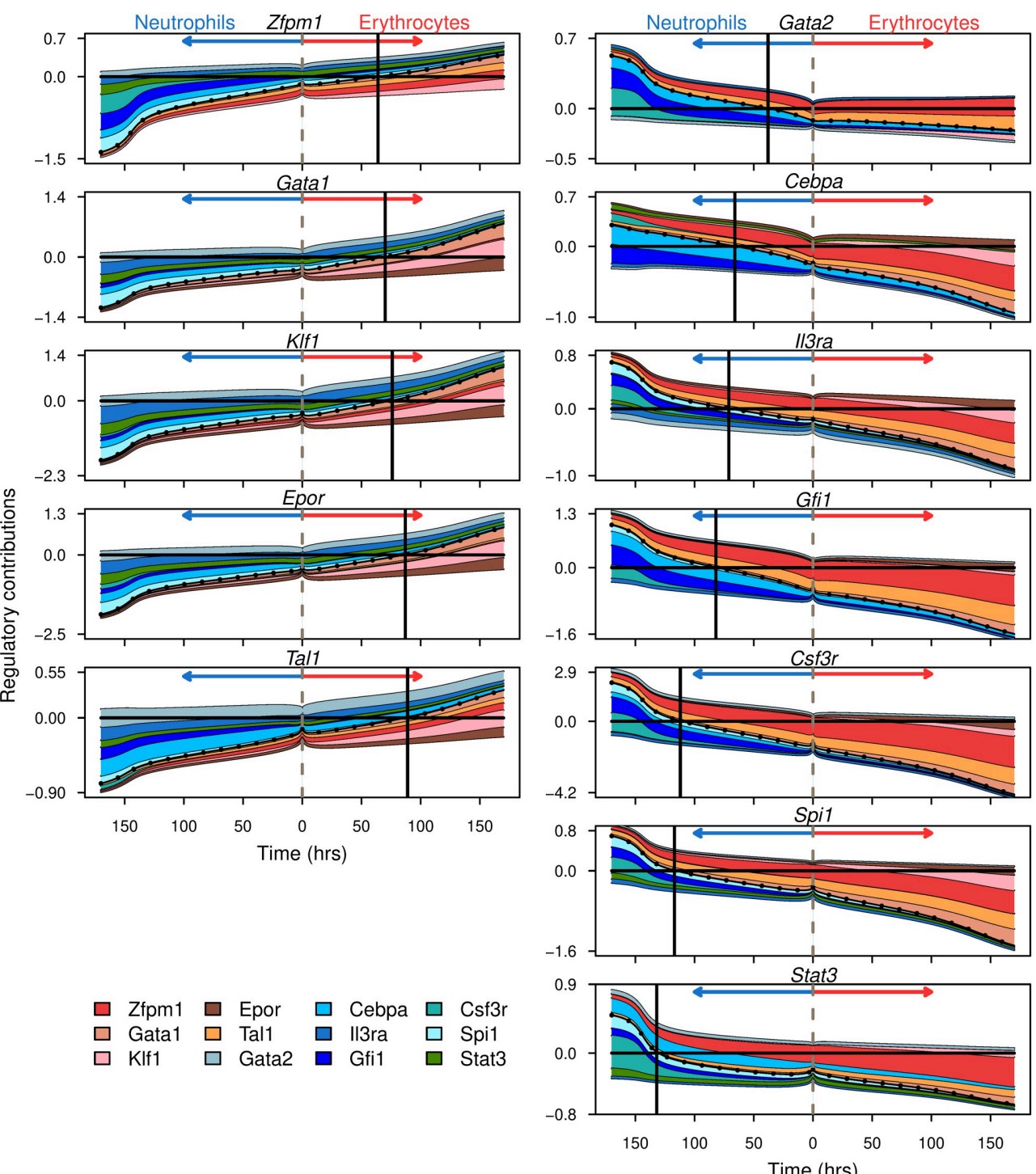

**Fig 6. The dynamics of gene regulation during differentiation.** The total regulatory input ($u$) is plotted as the dotted black line. The colored layers show the regulatory contribution of individual regulators. See Section 4 for the definitions of total regulatory input and regulatory contributions. The contributions of repressors and activators are shown above and below the dotted line respectively. The vertical dashed line in the center corresponds to uninduced FDCP-mix cells at the start of differentiation. Regulatory contributions during erythrocyte and neutrophil differentiation are shown to the right and left of the dashed line respectively. The vertical black line marks the time when the total regulatory input crosses zero so that synthesis occurs at half its maximum rate (Section 4).

differentiation process, while *Gata2* and *Cebpa* are the first ones to be activated. Unlike erythrocyte differentiation, during which three activators provided activation throughout the process, the genes accounting for activation change in time and with target gene.

PU.1 provides activation during the later stages of the differentiation once it has increased in expression. This is consistent with the observations that PU.1 acts primarily in a concentration-dependent manner [4, 44, 63] and that conditional *Spi1* knockout in adult bone marrow does not eliminate granulopoiesis but instead results in the development of immature granulocytes [64]. Csf3r also provides activation at late timepoints. *Cebpa* and *Gfi1* together account for most of the early activation of the genes upregulated during neutrophil differentiation in FDCP-mix cells (Fig 6). Although *Gfi1* expression is positively correlated with genes upregulated during neutrophil differentiation in FDCP-mix cells (Fig 1) and with *Spi1* in GMPs [53], Gfi1 is known to function primarily as a repressor in MPPs and the lymphoid and myeloid lineages [54, 56, 57, 65]. The activation role inferred here for Gfi1 during neutrophil differentiation could result from indirect regulation of its targets. Another factor is the high level of similarity between the expression of *Cebpa* and *Gfi1* in the training data (Fig 1) that renders the two factors interchangeable in the model. C/EBPα is known to directly activate itself, *Spi1*, *Csf3r*, and *Gfi1* during neutrophil differentiation (S2 Table; [39, 48–50, 66–69]). We conclude therefore that the activation of neutrophil targets by Gfi1 inferred by the model could, in fact, represent the activity of C/EBPα. Taken together this analysis implies that neutrophil development in FDCP-mix cells is driven by C/EBPα and potentially Gfi1 acting indirectly [70], which activate *Spi1* at later time points.

## 2.5 *Cebpa* and *Gfi1* expression precedes *Spi1* upregulation in the neutrophil lineage in mouse bone-marrow hematopoietic progenitor cells

We sought confirmation of the sequence of gene activation implied by our model of FDCP-mix cell differentiation in an independent experimental system. We analyzed Tusi *et al.*'s single-cell RNA-seq (scRNA-Seq) data from Kit$^+$ mouse bone-marrow HPCs [11]. Although scRNA-Seq data are a static snapshot of the progression of cell states during steady-state hematopoiesis, it is possible to infer the order of cell states under a few assumptions. Weinreb *et al.* developed Population Balance Analysis (PBA) [71], which computes the probability of transitions between the cell states—defined by genome-wide gene expression—observed in single-cell gene expression data and hence the probability that an intermediate cell state will evolve into some terminal cell fate (Fig 7A). Cell states corresponding to multipotential progenitors—the origin of the differentiation process—and committed unilineage progenitors—the termini of the differentiation process—are identified by the expression of marker genes. PBA assumes that there are no oscillations in cellular state so that the dynamics are governed by a potential function of cellular state and cells always move from higher to lower potential (Fig 7C; [71]). Under this assumption, it is possible to order the cells in developmental time by arranging them in order of decreasing potential (see [71] for details).

We profiled the expression of *Cebpa*, *Gfi1*, and *Spi1* in Tusi *et al.*'s dataset [11] by identifying cells having a high probability of becoming neutrophils based on the fate probabilities assigned to them by PBA (Fig 7A). The potential decreases with increasing neutrophil probability (Fig 7C) and it is possible to visualize how gene expression changes with developmental age at a single-cell level (Fig 7D) by following the direction of decreasing potential. Since single-cell read counts have considerable cell-to-cell variability, we also divided the potential into 11 bins containing an equal number of cells and averaged the expression over the cells in each bin (Fig 7B). *Spi1*, although expressed at lower levels at the earlier stages, changes relatively little until bin 6. *Spi1* is upregulated subsequently and reaches its maximum expression in bin 9

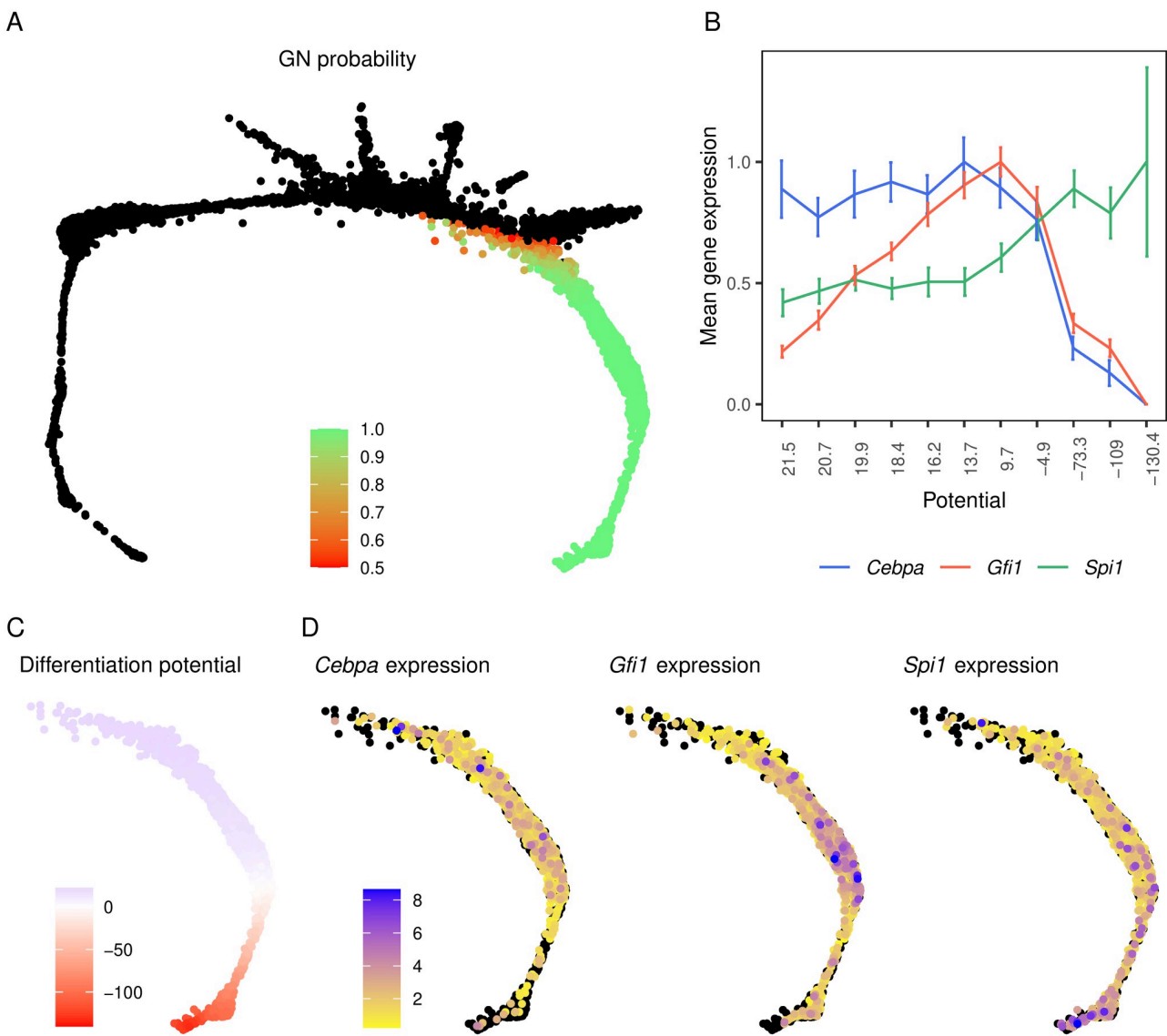

**Fig 7. The expression of *Cebpa*, *Gfi*, and *Spi1* in individual hematopoietic progenitor cells from murine bone marrow.** Panels **A**, **C**, and **D** are SPRING plots [72] of Tusi *et al.*'s scRNA-Seq dataset [11] of mouse bone-marrow derived Kit+ progenitors. Each point corresponds to an individual cell and cells are arranged as a k-nearest-neighbor (knn) graph according their pairwise distances in gene expression space [72]. **A**. The probability of a cell to adopt the neutrophil fate, as computed by the PBA algorithm, is shown as a color map if the probability is greater than 0.5. Cells with neutrophil probability less than 0.5 are shown as black dots. **B**. The mean expression of *Cebpa*, *Gfi1*, and *Spi1* in cells binned according to their potential (shown in panel C). Each bin contains 141 cells. The expression of each gene has been normalized relative to its maximum expression over the bins. The error bars show standard error. **C**. The potential landscape of the cells fated to be neutrophils is shown as a color map and orders the cells according to their maturity or developmental age. **D**. The expression of *Cebpa*, *Gfi1*, and *Spi1* is shown as a color map. Cells with no detected transcripts are plotted in black.

and maintains that level until the latest stage captured in this dataset. This temporal progression of *Spi1* expression is consistent with the patterns observed through live imaging of the PU.1 protein—it is expressed at low levels in HSCs and is upregulated during myeloid differentiation [12]—and the differentiation of FDCP-mix cells into neutrophils (Fig 1).

The scRNA-Seq data also show that *Cebpa* and *Gfi1* expression precedes the granulocyte-specific upregulation of *Spi1* (Fig 7B and 7D). *Cebpa* is already at its maximum level at the highest potential or earliest developmental stage. *Gfi1* rises rapidly at earlier stages and peaks

at bin 7. *Spi1* levels in bin 10 are greater than in bin 6 (*Cebpa* peak; Welch's one-sided two-sample t-test $p = 0.004$) or bin 7 (*Gfi1* peak; $p = 0.04$). Interestingly, both *Cebpa* and *Gfi1* are downregulated to lower levels in the latest developmental stages. These inferred temporal patterns of gene expression during the granulocytic differentiation of bone-marrow HPCs are consistent with our model's predictions that *Cebpa* and *Gfi1* are expressed earlier than and activate *Spi1* during neutrophil development.

## 3 Discussion

Despite our knowledge of the main genes effecting hematopoietic cell-fate decisions, their genetic architecture as well as the causality of their regulation is not fully understood. Here we have taken the approach, complementary to empirical genetic analyses, of learning the genetic architecture by training gene circuit models on gene expression time-series data. We trained a comprehensive model comprising 12 genes encoding TFs and cytokine signaling components on a high-temporal resolution dataset [25]. The correct predictions of the consequences of genetic perturbations at a quantitative level support the biological accuracy of the model. Similarly, we demonstrated through a detailed comparison with literature that the model correctly inferred the nature, activation or repression, of most known pairwise interactions. Our analysis implies that the genetic architecture of the erythrocyte-neutrophil decision is non-hierarchical and highly interconnected. There are extensive repressive interactions between genes from alternative lineages, while there is positive feedback from cytokine receptors. Furthermore, the gene circuit approach goes beyond static GRNs, and reveals their dynamics during the FDCP-mix cell differentiation process. We found that repressive interactions dominate at the earliest stages of the cell-fate decision while activation gains importance only at later stages. Finally, we show through model analysis followed by validation in an independent scRNA-seq dataset [11] that *Cebpa* and, possibly, *Gfi1* contribute to neutrophil development by upregulating *Spi1* and other downstream genes.

Hematopoietic cell-fate decisions have been modeled by two main approaches so far. In the first approach, the GRN is modeled using ODEs [3, 4, 73, 74] and the quantitative values of parameters are fixed by an exhaustive search of the parameter space to find regions that reproduce the qualitative behavior of the GRN. Such models have been mostly limited to 2–3 well-known "master" regulators, perhaps due to their relatively high computational expense. The second approach circumvents the high computational expense of ODEs by constructing logical or Boolean models that are more comprehensive and include 11–20 genes [75, 76]. The two approaches are similar in that the genetic architecture implemented by the models is based on prior empirical evidence.

The gene circuits built here differ from previous bistable-switch models in a number of ways. First, while bistable-switch models are constructed assuming a certain genetic architecture—mutual repression between two genes and autoactivation—gene circuits do not impose any interaction scheme beforehand but instead learn it from data. Gene circuits therefore offer an independent means of decoding the genetic architecture to supplement, but also to potentially refine, what we know from purely empirical approaches. The utility of this is illustrated by the fact that the gene circuits independently inferred the mutual antagonism between PU.1 and Gata1 and autoactivation of each gene that is baked into bistable-switch models, while diverging from them in also inferring that other factors, such as *Cebpa*, contribute to *Spi1* upregulation. Second, the gene circuits constructed here are more comprehensive, simulating a GRN of 12 genes compared to previous much smaller models [4, 25, 61] without resorting to Boolean networks that assume that gene expression is restricted to a few discrete levels. Third, while previous models and gene circuits differ in the precise

switch-like function employed, this difference is unlikely to matter since the parameter values are inferred by fitting.

Analysis of gene regulation dynamics in the model followed by validation in an independent dataset [11] led us to the insight that *Cebpa* and *Gfi1* are upregulated earlier than *Spi1* and drive the activation of *Spi1* and other neutrophil genes in FDCP-mix cells. *Spi1* has been thought to reside at the top of the hierarchy [2–4, 6, 25, 26] of white-blood cell genes since *Spi1* knockout mice lack all white-blood cells [38]. Additionally, evidence that PU.1 inhibits Gata1 [77] and vice versa [78] led to a model in which *Gata1* and *Spi1* form a bistable switch that decides the fate, while all the other genes are downstream targets of Gata1 or PU.1 [6]. However, the causal role of Gata1 and PU.1 in erythro-myeloid differentiation has been questioned recently by experiments in which Gata1 and PU.1 expression was monitored in differentiating HSPCs [12]. These experiments failed to detect an intermediate stage where cells co-expressed low amounts of both Gata1 and PU.1, which is a necessary condition for the fate decision to be driven by the genes' mutual repression. Furthermore, in cells destined for a myeloid fate, PU.1 was expressed at a constant level before being upregulated during the later stages of commitment while Gata1 remained undetectable throughout. This observation suggested that some factor or factors other than Gata1, unknown heretofore, drive PU.1 upregulation during myeloid differentiation. Our analysis therefore implicates *Cebpa* and, potentially, *Gfi1* as candidate upstream factors driving PU.1 upregulation during myeloid differentiation.

The activation of *Spi1* by *Cebpa* inferred here helps provide a link in the chain of causation leading to neutrophil maturation during FDCP-mix cell differentiation. The upregulation of *Spi1* is discernable only ∼50 hours after GCSF treatment and reaches its peak on day 7 (Fig 1), which is consistent with the pattern observed in mouse bone marrow (Fig 7). *Cebpa* is known to be upregulated by GCSF treatment [44, 49, 50, 79]. The C/EBPα protein is phosphorylated downstream of GCSF signaling [80] and autoactivates *Cebpa* transcription by binding to its promoter [68] and enhancers [48–50]. *Cebpa*, therefore, is a direct target of GCSF signaling, gets upregulated soon after GCSF treatment and activates *Spi1* subsequently. The late upregulation of *Spi1* could be reconciled with its mutant phenotype—the absence of all white-blood cells—if it were necessary for the activation of all white-blood cell genes, including those characteristic of neutrophil function. *Spi1* could then be seen as a hub which integrates input from lineage-specifying genes such as *Cebpa* and coordinates the expression of downstream functional genes.

Gene regulation during differentiation is dynamic; the contributions of the regulators modulating a gene's transcription and the overall balance of activation and repression change as the regulators' concentrations vary in time. Gene circuits, being dynamical models, allow us to determine how regulatory control varies in time both at the level of individual target genes (Fig 6) and more broadly at the network level (Fig 5). Our analysis indicates that, both at the individual and global levels, repression dominates over activation at earlier stages of eruthrocyte-neutrophil differentiation. As a result, all the genes in the network are partially repressed and expressed at low levels in progenitors. The data support this inference. Each gene in the network is upregulated by at least two-fold in one lineage or the other (Fig 1), which implies that the expression level observed in the progenitors is significantly below that of an actively transcribed gene.

The predominance of repression in the earlier stages implies, in turn, that the divergence of gene expression during differentiation is driven by relief of repression rather than by activation. This is similar to the idea of lineage priming [3, 4, 61, 81–83] in the bistable switch model [3, 4], where genes from alternative lineages are expressed at low levels and repress each others' expression in progenitors. Our model differs from the bistable switch model in two ways. First, whereas cell fate is selected by the initial concentrations of the two genes in the bistable switch

model, cytokines select the fate by exerting asymmetric effects on each gene in the gene circuits modeled here (Section 2). The second difference is that many more genes participate in cross-antagonism than Gata1 and PU.1 as hypothesized in the bistable switch model.

The overall balance shifts in favor of activation at later stages of differentiation, leading to the establishment of positive feedback loops between genes co-expressed in the same lineage. Of note is the activation of lineage-specific TFs by cytokine receptors. In the model, *Csf3r*, which codes for GCSFR, provides substantial activation to most of the genes upregulated in the neutrophil condition, while *Epor* performs a similar function in the erythrocyte condition (Fig 6). As discussed above, *Cebpa* is known to be downstream of GCSF signaling as are other myeloid TFs [40]. Similarly, EpoR phosphorylates and activates Gata1 through the PI3K-AKT pathway [84] and Epo signaling positively regulates several erythroid genes [85–88]. Cytokine receptor-mediated positive feedback has been shown to generate bistability in a model of Epo-dependent *Gata1* activation [41], resulting in greater sensitivity to Epo cytokine concentration. The positive feedback loops inferred in this bigger GRN might also result in bistability or multistability and sharp responses to cytokine concentration, a possibility that awaits confirmation through non-linear stability analysis [89].

Despite its general success in predicting the consequences of genetic perturbations, the model was unable to do so for *Gata2* knockdown (Fig 3) implying that *Gata2*-related inferences are incorrect for both FDCP-mix and *in vivo* differentiation. The model predicted nearly the exact opposite of the observed effects. The neutrophil lineage genes were predicted to be downregulated about two-fold, when in fact they were upregulated 1.2–4 fold, while erythrocyte lineage genes were predicted to be upregulated instead of being downregulated about two-fold (Fig 3). These mispredictions may be traced to the fact that Gata2-related parameters were not inferred with much certainty during fitting. 4 of 12 of the interconnectivity parameters ($T_{ij}$) where Gata2 is the regulator and 4 of 12 of the interconnectivity parameters where *Gata2* is the target are indistinguishable from zero among the gene circuits that met goodness-of-fit criteria (Fig 4 and S1 Table). This implies that the goodness-of-fit was insensitive to the type, activation or repression, of those interconnections. The uncertainty about how Gata2 regulates its targets and how it is regulated itself likely arises from the fact that there is almost no divergence in *Gata2* expression between the erythrocyte and neutrophil conditions (Fig 1), with differences discernible only at one time point out of thirty. The lack of different patterns of expression in the two conditions means that the *Gata2* data do not bear sufficient information to constrain *Gata2*'s regulatory parameters. Similarly, some of the inferences, such as the activation of *Spi1* and repression of *Gata1*, *Epor*, and *Klf1* by Gfi1 (Fig 4 and S2 Table), that did not match empirical data probably resulted from a lack of training data from MPPs, monocytes, and lymphocytic progenitors, where Gfi1 exerts the experimentally observed effects [58, 59, 70]. This limitation of the gene circuit methodology—that the training dataset may not contain sufficient information to accurately infer certain regulatory parameters—may be overcome by experimental designs that either sample differentiation trajectories in a larger number of conditions and cell types or after genetic perturbations.

In gene circuits, the interconnection between a pair of genes can represent both direct and indirect regulation of one by the other. Furthermore, gene circuits as implemented here do not include higher-order interactions such as the regulation of targets by a Fog1-Gata1 complex. These design choices have both advantages and disadvantages. On the one hand, this flexibility leads to inferred GRNs that are not completely specified mechanistically. We could not hope to delineate GRNs with biochemical details relying exclusively on gene circuits. On the other hand, this very flexibility also makes predictive modeling of GRN dynamics feasible. Although biochemically detailed models of intracellular signaling [41] and gene regulation [49, 50] have been constructed for individual pathways and enhancers, it is currently not possible to model

multiple signaling pathways or the gene regulation of multiple genes simultaneously. The challenges involved in constructing comprehensive but biochemically detailed models are many; the components are yet to be completely delineated, it is impractical to measure all the biochemical parameters, learning them from data leads to highly underdetermined problems, and the computational cost of such models would be prohibitive. Gene circuits, by coarse-graining much of the biochemical detail allow the construction of more complete models that are predictive in spite of a lack of biochemical detail.

The gene circuits derived here, being deterministic models trained on bulk gene expression data, are unable to account for stochasticity in gene expression or the effects of cellular heterogeneity in FDCP-mix populations. These limitations could potentially result in erroneous inferences of two types. First, the model may be overfit to the average initial conditions so that actual initial concentrations in single cells result in qualitatively different outcomes. Although we ensured that the inferred models were not fragile to errors of up to 70% in the mean expression of individual genes, it is possible that the inferred models produce non-biological outcomes in the presence of errors in the expression of multiple genes. Second, it is possible that the observed changes in averaged gene expression are not the result of gene expression modulation in single cells but that of changes in the sizes of phenotypically distinct subpopulations. Population heterogeneity could therefore lead to incorrect inferences about gene regulation. Most of the connections inferred here likely have a sound basis since a large proportion of them agreed with genetic and biochemical manipulations that are not confounded by cellular heterogeneity (Fig 4 and S2 Table). Furthermore, the handful of genes whose expression has been monitored live are clearly regulated at a single cell level [12–14]. Single-cell RNA-Seq data [11] also support the view that gene expression is changing at a single cell level and not as a result of varying proportions of admixed cellular subpopulations. This evidence does not rule out more complex scenarios where both single-cell and population-level processes contribute to the observed changes in mean gene expression and stochastic models trained on single-cell data would be necessary to uncouple these effects.

Our results show that the temporal dynamics of gene expression bear information about the genetic architecture underlying cell-fate choice. With a few exceptions such as the segmentation system of *Drosophila* [90], our current knowledge of the genetic architecture of most developmental systems is based on genetic analyses carried out at end points. Coupling gene circuits with high temporal resolution time series data is a viable complementary approach to decode the genetic architecture and reveal the causality of events during differentiation. One potential drawback of this approach is the cost of sequencing. However, the cost of sequencing is expected to decline exponentially over time [91] and is not likely to be a limitation in the future. Another concern is the high computational cost of fitting the gene circuits, which entails the use of parallel computers. This challenge was recently overcome by an algorithm called Fast Inference of Gene Regulation (FIGR) [16] that is much more computationally efficient and can infer models on a consumer-grade computer in a reasonable amount of time. We anticipate that with these improvements, it will be possible to collect time series datasets that span multiple hematopoietic lineages and genetic backgrounds and use the gene circuit approach to comprehensively decode the genetic architecture of hematopoietic cell-fate decisions.

## 4 Materials and methods

### 4.1 Gene circuit model of Erythroid-Neutrophil differentiation

The initial conditions were given by the mRNA concentrations in progenitor cells. Eq 1 were solved numerically using the Bulirsch-Stöer adaptive step-size solver to an accuracy of $10^{-3}$ as described previously [23].

### 4.2 Training data

The gene circuit was trained on May *et al.*'s genome-wide gene expression time-series dataset (GEO GSE49991; [25]) acquired during the differentiation of FDCP-mix cells into erythrocytes or neutrophils. See [25] for the details of data processing and cross-sample normalization. The expression level of each gene was further normalized against its maximum expression in either condition for model training and visualization.

### 4.3 Optimization by parallel lam simulated annealing (PLSA)

The parameters of Eq 1 were inferred by minimizing the cost function

$$E = \sum_{i,m,l} (x_i^l(t_m) - \hat{x}_i^l(t_m))^2 + \text{Penalty}, \tag{2}$$

where $x_i^l(t_m)$ and $\hat{x}_i^l(t_m)$ are model output and data respectively for gene $i$ in lineage/condition $l$ at time $t_m$. The penalty is a weighted regularization term that limits the search space or magnitude of the regulatory parameters $T_{ij}$, $b_i$, and $h_i$. The penalty is given by

$$\text{Penalty} = \begin{cases} \exp(\Pi) - \exp(1), & \text{if } \Pi > 1 \\ 0, & \text{otherwise}, \end{cases} \tag{3}$$

where

$$\Pi = \sum_i \Lambda_i \left( \sum_j (T_{ij}\hat{x}_j^{\max})^2 + (c^{\max})^2 + h_i^2 \right).$$

$\hat{x}_j^{\max}$ is the maximum expression of gene $j$ observed in the dataset [92] and $c^{\max}$ is the maximum value of $c^l$ over all conditions $l$. $\Lambda_i$ controls the magnitude of the regulatory parameters of gene $i$. $\Lambda_i$ was set to 0.1 for all genes except *Csf3r*, for which $\Lambda_i$ was set to 0.01. This allowed *Csf3r*'s regulatory parameters to have larger values, which was necessary for the model to be able to recapitulate the large dynamic range of *Csf3r* expression data (Fig 1).

The cost function (Eq 2) was minimized using parallel Lam simulated annealing (PLSA)—simulated annealing with the Lam cooling schedule [93]—running in parallel [17] as described previously [22]. PLSA was carried out on 10 CPUs (Intel Xeon E5–2643 v3 cores) in parallel.

### 4.4 Selection of gene circuits for analysis

Since PLSA is a stochastic method [17], each optimization attempt results in different values of inferred parameters and hence in a distinct gene circuit model. In order to evaluate their reproducibility, we repeated the optimization to obtain 100 different gene circuits. The root mean square (RMS) score,

$$\text{RMS} = \sqrt{\frac{E}{N_d}}, \tag{4}$$

where $N_d$ is the total number of data points, was used to measure the goodness-of-fit of each gene circuit model. We chose 71 gene circuits having RMS scores lower than 0.06, corresponding to an average error of 6% in expression levels. Models with higher RMS scores showed qualitative defects in their expression patterns compared to data.

### 4.5 Significance of fits

The optimization problem for the 12-gene circuit is overdetermined, having 720 data points and 192 free parameters, and the risk of overfitting is minimal. Nevertheless, we checked whether the model fits captured temporal patterns inherent in the data or whether the degrees of freedom were so numerous that the model could fit randomized non-biological data equally well. We randomized the data in a manner that preserved the dynamic range of the real data while creating non-biological temporal expression patterns and tested the ability of gene circuits to fit the latter compared to the former. For each gene, we created chimerical temporal expression patterns by combining erythrocyte training data up to the 96 hour time point with neutrophil data at later time points and vice versa. In each synthetic dataset, 10 of 12 genes were given chimerical expression patterns while the other two retained the original training data. 66 such synthetic datasets were generated for each combination of 10 genes (Algorithm 1). 10 gene circuits were trained per dataset resulting in a total of 660 gene circuits. The RMS scores of the resultant gene circuits were compared to the 100 gene circuits trained on the real data. The statistical significance of the differences between the RMS scores of gene circuits trained on random and real data was determined using the Wilcoxon rank sum test with continuity correction.

**Algorithm 1** Gene expression swapping between two conditions

```
1: Cᵢ: combination i for 10 out of 12 genes
2: cᵢ: 2 genes not included in Cᵢ
3: time_points: 30 sampled differentiation time points, time_points ∈
{0, 2, 4, ..., 96, 120, 168}
4: x_ery(i, g, t) ← gene expressions in erythrocyte condition for gene
combination i, gene g, and time point t
5: x_neu(i, g, t) ← gene expressions in neutrophil condition for gene
combination i, gene g, and time point t
6: x_eryW(i, g, t): empty 66 × 12 × 30 array for storing swapped and nor-
mal gene expressions in erythrocyte condition for gene combination i,
gene g, and time point t
7: x_neuW(i, g, t): empty 66 × 12 × 30 array for storing swapped and nor-
mal gene expressions in neutrophil condition for gene combination i,
gene g, and time point t
8: fileᵢ: output file for writing swapped and normal expressions for
gene combination i
9: for i in 1:¹²C₁₀ do
10:    for gene g in Cᵢ do
11:       for t in time_points do
12:          if t < 96 then
13:             x_eryW(i, g, t)←x_ery(i, g, t)
14:             x_neuW(i, g, t)←x_neu(i, g, t)
15:          else
16:             x_eryW(i, g, t)←x_neu(i, g, t)
17:             x_neuW(i, g, t)←x_ery(i, g, t)
18:          end if
19:       end for
20:    end for
21:    for gene g in cᵢ do
22:       for t in time_points do
23:          x_eryW(i, g, t)←x_ery(i, g, t)
24:          x_neutW(i, g, t)←x_neu(i, g, t)
25:       end for
26:    end for
27:    for gene g in (Cᵢ and cᵢ) do
28:       for t in time_points do
```

```
29:        WRITE(file_i, x_eryW(i, g, t), x_neuW(i, g, t))
30:     end for
31:   end for
32: end for
```

### 4.6 The sensitivity of the model to initial conditions

The initial concentration of each gene was perturbed by ±10, 20, 30, 50, 70%, one gene at a time. Model 66 was run with the perturbed initial conditions (120 simulations) and the RMS for each simulation was calculated.

### 4.7 Simulation of perturbation experiments

*Gata1* and *Spi1* knockout was simulated by setting their initial concentrations and mRNA synthesis rates $R_i$ to zero.

To simulate the knockdown and overexpression experiments carried out by [25] in FDCP-mix cells, we chose one representative model from the 71 that had met the goodness-of-fit criteria. For each model, we determined the number of regulatory parameters ($T_{ij}$) that had the same sign as the majority of the models. Of the 7 models having the largest number of regulatory parameters aligning with the consensus, one model, model #66, was chosen for perturbation simulations.

The knockdown *Spi1* or *Gata2* in FDCP-mix cells was simulated by decreasing the maximum synthesis rate of the gene, $R_{Spi1}$ or $R_{Gata2}$, respectively. Since the efficiency of the knockdown achieved in the specific experiments was unknown, we chose the value of $R_{Spi1}$ or $R_{Gata2}$ so that the simulated expression of *Spi1* or *Gata2* matched the empirical values respectively. The simulations therefore could be said to predict the expression of only 11 of the 12 genes.

In the PU.1ERT and GATA1ERT experiments, 4-hydroxy-tamoxifen (OHT) treatment did not directly modulate the amount of *Spi1* or *Gata1* mRNA but instead increased the activity of the constitutively expressed PU.1ERT and GATA1ERT fusion proteins. We simulated the increase in the activity of PU.1 or Gata1 by introducing a constant bias term $B_i$ in the total regulatory input $u$ of each gene $i$,

$$u = \sum_{j=1}^{N} T_{ij} x_j^l + b_i c^l + h_i + B_i.$$

The bias term is proportional to the genetic interconnectivity parameter corresponding to the regulation of each gene by PU.1 or Gata1 so that $B_i = T_{i \leftarrow Spi1} \cdot \beta_{Spi1}$ or $B_i = T_{i \leftarrow Gata1} \cdot \beta_{Gata1}$ respectively. The proportionality constants $\beta_{Spi1}$ and $\beta_{Gata1}$ represent the additional amount of active PU.1 and Gata1 induced by OHT respectively. Similar to the knockdown experiments, the efficiency of activation achieved in the overexpression experiments was unknown and we chose the values of the proportionality constants to match the observed expression of 1 of 12 genes. We did not however fit to the observed expression of the overexpressed gene since it stems from a mixture of mRNAs transcribed from the endogenous locus and the constitutively expressed ERT fusion gene. Instead we chose the values of the proportionality constants so the simulations matched the observed expression of *Gata1* in the PU.1ERT and *Spi1* in the GATA1ERT experiments respectively.

The simulations were carried out with $c^l = 0$ to simulate the progenitor condition since the experiments had been conducted in undifferentiated FDCP-mix cells. The simulations were compared to experimental data at equilibrium. The GRN was simulated for 1000 hours to allow the solution to reach equilibrium. The ratio of each gene's expression in the perturbed condition to its expression in the unperturbed condition was computed to determine the fold

change predicted by the simulation. This was compared to the empirical fold change, computed as the ratio of gene expression in treated cells to gene expression in control cells.

## 4.8 Analysis of gene regulation dynamics

The contribution of individual regulators to the activation or repression of a target was determined by decomposing the total regulatory input $u = \sum_{j=1}^{N} T_{ij} x_j^l + b_i c^l + h_i$ into its individual terms. The contribution of regulator $j$ to the regulation of gene $i$ was determined by computing $T_{ij} x_j^l(t)$, where $T_{ij}$ is the genetic interconnectivity of the two genes and $x_j^l(t)$ is the model solution for the mRNA concentration of gene $j$ at time $t$ and condition $l$. Since the mRNA concentrations vary in time, the relative contributions of the regulators to the activation or repression of any target also vary in time. When the total regulatory input crosses 0, that is $u = 0$, the regulation-expression $S(u) = \frac{1}{2}\left(u/\sqrt{(u^2 + 1)} + 1\right) = \frac{1}{2}$ and the mRNA is synthesized at half the maximum rate (Eq 1). The time at which different genes achieve half-maximum expression was used to order their activation in time.

## 4.9 Visualization of Tusi *et al.*'s scRNA-Seq data [11]

The expression of *Cebpa*, *Gfi1*, and *Spi1* in individual Kit$^+$ hematopoietic progenitors cells from mouse bone marrow (GEO GSE49991; [11]) was visualized as follows. The cells were arranged in 2D space as a k-nearest-neighbor (knn) graph according to their pairwise distances in gene expression space (SPRING algorithm; [72]). The potential landscape and the probability of each cell to adopt a given fate were given by Population Balance Analysis (PBA; see [71] for details). Genome-wide normalized gene expression counts, the PBA potential, the PBA lineage probability, and the 2D SPRING coordinates of each cell were obtained from https://kleintools.hms.harvard.edu/paper_websites/tusi_et_al/.

## Supporting information

**S1 Fig. Expression of the modeled genes in the Tusi *et al*. scRNA-Seq dataset [11].** The average expression in MPPs, erythroid progenitors, and granulocytic progenitors is shown for the modeled genes. Erythroid and granulocytic progenitors were identified as having a PBA erythroid and granulocytic probability (see Materials and methods) greater than 0.9 respectively. MPPs were identified as cells having a low PBA probability ($< 0.2$) of belonging to any lineage. Error bars show the standard error of the mean.
(PDF)

**S2 Fig. The significance of gene circuit fits.** The distributions of the RMS scores of gene circuits trained on real data (Unpermuted models) or on randomized synthetic data (Permuted models) are shown as violin plots. The scores were compared using the Wilcoxon ranksum test with continuity correction ($p = 3.8 \times 10^{-8}$).
(PDF)

**S3 Fig. Sensitivity of the model to initial conditions.** Model 66 was run with the initial conditions perturbed one gene at a time. The *x*-axis is the magnitude of the perturbation. The *y*-axis is the RMS. The perturbed gene is indicated by the color of the points. The dotted line is the RMS of model 66 and the black line is the goodness-of-fit threshold RMS.
(PDF)

**S4 Fig. Simulation of *Gata1* knockout.** *Gata1* knockout was simulated in all 71 models that met the goodness-of-fit criteria. Their output is plotted as lines. The symbols and colors are

the same as [Fig 1].
(PDF)

**S1 Table. The values of the parameters of the gene circuit models that met the goodness-of-fit criteria.** Columns correspond to parameters while rows correspond to models. $T_{ij}$ are shown as T_Gene$i$_Gene$j$, $b_i$ are shown as b_Gene$i$, $h_i$ are shown as h_Gene$i$, $R_i$ are shown as R_Gene$i$, and $\lambda_i$ are shown as lambda_Gene$i$.
(TSV)

**S2 Table. Comparison of model predictions with published experimental evidence.** Each row compares a model prediction about a genetic interconnectivity parameter $T_{ij}$, representing the regulation of gene $i$ by gene $j$, with published experimental evidence. Comparisons of the same parameter to multiple papers are listed in separate rows. $T_{ij}$ is listed as T_Gene$i$_Gene$j$. The prediction column lists the type of regulation inferred by the model. It shows activation or repression when the first quartile of the distribution of the inferred parameter is positive or if the third quartile of the distribution is negative respectively ([Fig 4]). The prediction column shows "sign not constrained" when the interquartile range spans negative and positive values. The experiment column lists that type of interaction established in the paper. If the paper describes evidence only of binding but not whether the target is activated or repressed, then the entry is "binding". Negative experimental results are listed as "no effect found". The entry in the experiment column is "not found" if we were not able to find any published tests of the parameter in question. The "Status of prediction" columns lists whether the evidence matches the prediction or not. "confirmed" implies agreements, while "incorrect prediction" implies disagreement. An asterisk indicates that conflicting experimental evidence was found. Conflicting evidence was found for the regulation of *Gata1*, *Spi1*, and *Gata2* by Gfi1 (Moignard *et al*. [53]). Situations where no evidence was found or the paper reported negative results are listed as "undetermined". The "Type of evidence" column classifies the evidence as genetic, protein-protein interaction, *cis* regulation, or functional *cis* regulation. Genetic evidence involves genetic manipulation of the predicted regulator followed by a characterization of the target's expression and usually cannot distinguish between direct and indirect effects. Protein-protein interaction implies biochemical evidence of direct protein-protein interactions. *cis* regulatory evidence indicates direct interactions by identifying regulatory elements or binding sites potentially bound by the predicted regulator but does not establish a functional relationship between binding and the expression of the target gene. Functional *cis* regulation goes a step further and manipulates the binding sites and measures reporter or target expression to provide evidence that the binding of the regulator has functional impacts. The "Organism/Cells" and "Citation" columns list the organism or cells in which the interaction was tested and the DOI URL for the paper respectively.
(TSV)

## Acknowledgments

We thank YL Loh for discussions and comments.

## Author Contributions

**Conceptualization:** Joanna E. Handzlik, Manu.

**Funding acquisition:** Manu.

**Investigation:** Joanna E. Handzlik.

**Methodology:** Joanna E. Handzlik.

**Software:** Joanna E. Handzlik.

**Supervision:** Manu.

**Validation:** Joanna E. Handzlik.

**Visualization:** Joanna E. Handzlik.

**Writing – original draft:** Joanna E. Handzlik, Manu.

**Writing – review & editing:** Joanna E. Handzlik, Manu.

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
