## [Decision Letter · Decision Letter 0]

7 Jul 2021

Dear Dr. Manu,

Thank you very much for submitting your manuscript "Data-driven modeling predicts gene regulatory network dynamics during the differentiation of multipotential hematopoietic progenitors" for consideration at PLOS Computational Biology.

As with all papers reviewed by the journal, your manuscript was reviewed by members of the editorial board and by several independent reviewers. In light of the reviews (below this email), we would like to invite the resubmission of a significantly-revised version that takes into account the reviewers' comments.

We cannot make any decision about publication until we have seen the revised manuscript and your response to the reviewers' comments. Your revised manuscript is also likely to be sent to reviewers for further evaluation.

Sincerely,

Stuart A Newman

Guest Editor

PLOS Computational Biology

Ilya Ioshikhes

Deputy Editor

PLOS Computational Biology

Reviewer's Responses to Questions

**Comments to the Authors:**

Reviewer #1: The authors use an ODE model framework to predict the dynamics of a bipotent hematopoietic progenitor cell fate choice and compute the pairwise regulatory interactions driving the process, based on a network of 12 regulators. They generate 71 models that match their quality control criteria and then describe in depth the performance and predictions from one of these models. In many ways, this model performs very well relative to experimental data. It is a tour de force, likely to provide an influential, encouraging precedent for other gene network modeling attempts.

The main drawback of the submitted manuscript arises from the fact that it is focused on a question that has been studied in depth with a variety of experimental as well as computational tools for over 20 years. Thus, one issue is that major biological conclusions of this paper are less novel than the wording suggests in some places. The key importance of Gfi1 and Cebpa for neutrophil differentiation has been demonstrated by multiple groups, and PU.1’s demonstrated roles in macrophage, dendritic cell, and lymphocyte development would arguably make it less likely to be the master regulator of neutrophil identity in any case. Another problem is that although the scholarship in the paper is excellent, it still misses a few relevant pieces of evidence from the literature that should have had some impact on the modeling. The reader is left wondering whether the chosen model really is the one that would have performed the best, if these other results were taken more fully into account. Thus, it would be good for the authors to address the following points more directly in the manuscript.

1. It is somewhat troubling that the model is framed as though the normal hematopoietic decision is between erythrocyte fate and granulocyte fate. That is clearly not a direct binary choice in the Tusi et al 2018 data used in the later part of the paper, where the non-erythroid branch leads to sub-branches for lymphocytes, dendritic cells, and macrophages before the long neutrophil spur emerges. The granulocyte-macrophage decision network has been dissected at the single-cell level by Olsson...Grimes 2016, for example. Even with FDCP-mix cells, the original Spooncer 1986 paper shows macrophage emergence as well as erythroid and neutrophil production. Did any of the 70/71 models that were not shown in depth in the paper incorporate more alternative lineage options?

2. The modeling is based on an underlying picture that the cell population moves uniformly in responding to transcription factor circuits. However, Tusi et al 2018 and many other recent studies have emphasized that single cells in such differentiating populations behave heterogeneously. While one could average these population variances and assume that they are just noise within a homogeneous population response, it is also possible that the true gene network genuinely supports more complex lineage choices, with some “built-in” gene circuit responses leading to other fate branches. I think this possibility is important for the authors to address more explicitly, for two reasons.

a) First, in this model, it’s assumed that if two transcription factors are both expressed at certain levels in the population, then they are expressed in the same cells. However, they may not be – the Gata2 expression may be in different cells from the Gfi1 expression, for example. While a different kind of model would be needed to deal with probabilistic aspects of heterogeneous population responses, it would be good to acknowledge that this is an issue. Ideally, the authors might calculate how robust their top models would be in a few examples where the population was generating some reasonable, specific mixtures as intermediates.

b) Second, related to point 1, the goal here is to infer the true underlying gene network structure on the basis of the gene expression signatures that predominate. But some gene network features might be missed if they led the cells to dead ends in the culture systems used by May et al. Are erythroid genes unexpressed in the neutrophil conditions because they are actively repressed, or because any cells taking the erythroid pathway are dying out? Also, if Gata1-nonexpressing cells were to upregulate PU.1 prematurely, would this drive them toward a macrophage fate that makes them proliferate less, in both of the tested sets of culture conditions? Is this an alternative explanation for why earlier PU.1 upregulation is not seen? It would be helpful if the authors discussed the possibility that pathway-specific cell-survival constraints could also be “trimming” the outputs from the gene network. Would this reduce the need for direct repression operating early in the network?

3. Please, be clearer about how the model takes into account possible dose-dependent changes in regulator impact. This is important, since there is much published evidence for different developmental effects of PU.1, at least, depending on its level of activity, either absolutely or relative to other factors. One possibility is that these factors always act in the same direction on a given target, and in a continuous-valued way as a function of dosage. Another is that these factors always act in the same direction on the target, but have a sharp dosage threshold below which they have no effect. A third possibility would be that a factor might switch its effect from positive at a low level of expression to inhibitory at a high level of expression (e.g. by binding to different sites as its concentration rises). How many of these possibilities are taken into account in the model construction?

4. It was surprising not to see more explicit use of evidence for the relationship between Gata1 and Gata2. The Zfpm1-dependent “Gata switch” (Gata2+Zfpm1 first activating Gata1, but Gata1+Zfpm1 then repressing Gata2) is considerably discussed in the literature (Orkin and Bresnick groups) but does not seem to emerge from the model. Could the lack of the Gata switch explain some of the problems with predicting Gata2 levels?

5. Fig. 5 does not seem show any evidence for transcription factors acting obligatorily in pairwise combinations in order to generate specific outcomes. The structure of equation 1 seems to require measuring each factor’s effect on the other genes alone, one factor at a time. One of the major activities in the early stages of the erythroid pathway in Fig. 5 seems to be Zfpm1 by itself. However, Zfpm1 presumably cannot bind DNA by itself. Also, there is considerable evidence that Gata factors with Zfpm1 function differently from Gata factors without Zfpm1 (e.g. work from Crispino). Does the model take into account the inability of Zfpm1 to affect gene expression unless it is binding to a Gata factor?

6. In Table S1, the headings are misaligned with the columns of parameter values – please check and fix!

Minor:

1. The conclusion that cells can go far in the neutrophil lineage with minimal PU.1 expression fits the evidence that complete PU.1 knockouts do generate immature granulocyte-like cells, but not lymphocytes or macrophages (Dakic...Nutt 2005). This experimental observation is in good harmony with the predictions of the model and should be noted.

2. There is evidence in the literature using sensitive single-molecule detection that PU.1 and GATA-2 are very frequently coexpressed in individual myelo-erythroid progenitors (Wheat et al 2020 Nature). Therefore, it is misleading to imply, as on p. 3 and p. 19, that PU.1 is not expressed in granulocyte-lineage differentiation at all until its late increase in expression.

3. The references should be re-checked. On p. 9, is the citation to ref 45 as meant? Also, there are some references duplicated under different numbers: 39 the same as 67, 44 the same as 63.

4. Several times throughout the manuscript, “its” (meaning, belonging to it) is spelled “it’s”.

5. Please provide tick marks for the right-hand plots in Fig. 2. Are these “relative expression” units the same as the ones in Fig. 1?

Reviewer #2: In this manuscript, gene circuits are trained on high resolution temporal gene expression data across two differentiation pathways, toward erythroid cells and toward neutrophils, from a mouse cell line with the potential to differentiate along multiple lineages (FDCP-mix). Twelve genes implicated previously in these differentiating lineages, both transcription factors and signaling components, comprise a "gene regulatory network" (GRN) that is the focus of the paper. The resulting gene circuits are dynamic models from which the type (negative or positive) and magnitude of regulatory interactions can be inferred (which the authors call the genetic architecture). Further analysis provides inferences as how the regulatory contributions of each of the members of the GRN change over the courses of the two differentiation lineages. Tests of the predictions of the gene circuit models were confirmed for some but not all cases. A major conclusion with potential for substantial biological impact is that the protein PU.1, encoded by the Spi1 gene, is not the major early regulator of differentiation toward neutrophils, but rather it functions downstream of early regulator(s) Cebpa and (perhaps) Gfi1.

This manuscript has potential to provide an approach, specifically gene circuit analysis, that is not commonly utilized in analyzing GRNs. It also has the potential to bring important new insight into an intensively studied field, i.e. the TFs and signaling molecules that are primary determinants of cell fate decisions. However, the manuscript will need to be improved in several ways to achieve those potentials.

(1) The construction and products of the gene circuits need to be explained specifically and clearly. It is not clear how the ordinary differential equations (ODEs) obtained by training gene circuits differ from those that led to the bistable switch models espoused by May, Enver and others. The manuscript does state that prior work with ODEs was limited to a small number of genes. What are the fundamental differences between the described and prior work? Exactly what is the output of the gene circuit analysis? A table of interaction coefficients is provided, but it seems that many other parameters are modeled. Can the resulting models be used by others in further analyses?

(2) What was the logic for including only the 12 genes in the circuits and networks? Multiple lines of investigation support important roles for the 12 genes in blood cell differentiation, but those prior studies also implicate other genes. For instance, roles for Gfi1b, Meis1, Fli1 (and genes encoding other ETS factors in addition to PU.1), Runx1, other genes. Was there a computational limit on the number of genes that could be included? Were the major observations (e.g. highly interconnected networks, inferred late role for Spi1) robust to the inclusion of more genes or replacement with other genes?

(3) While FDCP-mix cells have been useful for several studies, they are not the same as primary, multi-lineage progenitor cells. Thus, it would be wise to insure that the training data used (from the May et al. study in FDCP-mix cells) are consonant with data from primary cells. For example, the data summarized in Fig. 1 show RNA levels for Gata2 increasing during neutrophil differentiation, but my examination of multiple RNA-seq datasets across hematopoietic differentiation does not reveal that accumulation of GATA2 in primary neutrophils or monocytes.

(4) The manuscript states the bistable switch model in the simplest terms, but in the May et al. paper (that is the source of the RNA-seq data used in this manuscript)s, the model has evolved to a three-way model with GATA1, GATA2, and PU.1. Importantly, the May et al. paper has ChIP-seq data that supports co-occupancy by GATA2 and PU.1 in the "multilineage progenitor cells". The manuscript under review should evaluate this more advanced, 3-protein model by the gene circuits.

(5) The manuscript includes an analysis of single cell RNA-seq to evaluate their major conclusions about determinants of neutrophil cell fate. However, it is not clear that those data argue against the bistable switch model with Spi1 (PU.1) playing an early role. The single cell RNA data shows that the Spi1 transcript levels reach a peak after the peaks of Cebpa and Gfi1 (Fig. 7), supporting the conclusions in the manuscript. However, those data also show that Spi1 transcripts are present, even higher than Gfi1, in the early stages. The fact that Spi1 transcripts are present does support the frequently stated binary (bistable switch) model for antagonistic repression between PU.1 (SPI1) and GATA1.

(6) Text in the Discussion implicates GCSF signaling as a key instruction for differentiation to granulocytes. The gene encoding the receptor for GSCF, CSF3R, is part of the GRN. However, its expression rises dramatically late in the differentiation to neutrophils. This is the same kind of data that the authors use to argue that the Spi1 gene cannot be an early determinant of differentiation. The manuscript should explain this apparent discrepancy and utilize the same criteria for inference about early versus late-acting factors.

(7) The text is overly long, and it needs to be streamlined.

(8) The manuscript needs to state some of the conclusions more carefully. For example, on page 20: "We showed that this model is biologically accurate by correctly predicting the consequences of genetic perturbations..." would be more appropriately stated as "The correct predictions of the consequences of genetic perturbations... support the accuracy of the model". At the same time, the authors should state how the inaccurate predictions of the model should be evaluated in terms of overall utility of the models.

(9) Fig. 1: What are the units for the levels of RNA? Are they z-scores?

(10) In Fig. 7, panels B, C, and D appear to be switched relative to the descriptions in the legend.

(11) The link (p. 32) to the SPRING coordinates of cells in the Tusi et al. paper appears to be broken.

**Have the authors made all data and (if applicable) computational code underlying the findings in their manuscript fully available?**

Reviewer #1: Yes

Reviewer #2: **No: **I did not see any comment in the manuscript about availability of software or code.

PLOS authors have the option to publish the peer review history of their article (what does this mean?). If published, this will include your full peer review and any attached files.

Reviewer #1: No

Reviewer #2: No
---

## [Decision Letter · Decision Letter 1]

23 Oct 2021

Dear Dr. Manu,

Thank you very much for submitting your manuscript "Data-driven modeling predicts gene regulatory network dynamics during the differentiation of multipotential hematopoietic progenitors" for consideration at PLOS Computational Biology.

As with all papers reviewed by the journal, your manuscript was reviewed by members of the editorial board and by several independent reviewers. In light of the reviews (below this email), we would like to invite the resubmission of a significantly-revised version that takes into account the reviewers' comments.

We cannot make any decision about publication until we have seen the revised manuscript and your response to the reviewers' comments. Your revised manuscript is also likely to be sent to reviewers for further evaluation.

Sincerely,

Stuart A Newman

Guest Editor

PLOS Computational Biology

Ilya Ioshikhes

Deputy Editor

PLOS Computational Biology

Reviewer's Responses to Questions

**Comments to the Authors:**

Reviewer #1: Handzlik and Manu have carefully responded to the previous critiques, and their model represents a valuable contribution to the field. Their programme to use time-series data as the basis for gene network modeling is clearly an important corrective to gene network modeling based only on static correlations, and so it is hoped that this work will be influential. The changes in the manuscript in response to the previous reviews definitely strengthen it and make it clearer in its implications. They have done a good job of dealing with the problems of non-physiological Gata2 regulation in the FDCP-mix cells from which the time course data were drawn. Still, there are some problems or obscurities in this version of the manuscript that could mislead the reader.

1. In previous reviewer comments, there was some concern about the interpretation of PU.1 effects, based on the different dose-dependent impacts of PU.1. In general, lower levels of PU.1 seem to be important in B cell development and neutrophil development, while higher levels of PU.1 drive macrophage/monocyte development. The authors have added several comments acknowledging that PU.1 effects are dose-dependent, but they do not really explore what this entails. In fact, this problem is also related to another reviewer concern, which was the neglect of any other fate choices that the cells may face besides the one between neutrophil and erythroid fates.

This concern is directly relevant to the model predictions as compared to measurements shown here. Whereas B cell development is not accessible from the FDCP-mix model, there is good published evidence that the neutrophil (low PU.1) vs. macrophage (high PU.1) fate choices involve genuine mutual repressions between alternative sets of PU.1 collaborators, C/EBPalpha and Gfi1 on the one hand, and Irf8, potentially also Egr1 and 2, on the other hand (Laslo et al. 2006 Cell, Olsson et al 2016 Nature). Irf8 and the Egr factors are not included in the model, but it is at least possible that they are causing the otherwise paradoxical antagonisms seen when PU.1 levels are raised abnormally high, in the PU.1-ERT data in Fig. 3. If this explanation is correct, then the “conflict” between data and model predictions arises only from the simplifying assumptions in the model that disregard mechanisms involved in fate branches other than the neutrophil/ erythrocyte one.

Obviously the authors should not have to make a completely new model at this point, especially when there is no comparable time series evidence to use for optimization. But it is worth making it clearer that the dose dependences of factor actions (presumably for Gfi1 as well as for PU.1) can lead to alternative pathways, and that the regulators of these alternative pathways are not all included in the model. Also, the data showing PU.1-ERT activation shutting down the neutrophil program is not an anomaly, but exactly what would have been predicted based on evidence from Laslo et al., 2006 (Cell), and Kueh et al., 2013 (Science), although those were not whole-genome studies.

2. The choice of baseline values in the modeling is somewhat surprising. As noted in passing in the text (section 2.4), the model (section 4.1) structures the sigmoidal regulation-expression level function S(u) such that when regulatory input is zero, the value of S is 0.5. Since values <0 are allowed in the model, it is not clear why baseline transcription is arbitrarily set at half maximal.

a. This is a peculiar choice for a mammalian system, where most would consider that the default for a gene with zero regulatory input would be no expression. Please explain.

b. It also leads the authors to interpret values of expression between 0 and 0.5 x maximal (in Fig. 6, at time 0) as evidence that genes are actually undergoing repression (p. 17), whereas most would interpret values >0 as evidence of weak activation. The authors’ interpretation of multilineage priming thus becomes the exact opposite of what others in the field have thought, even though based on the same data.

3. The validation of late Spi1 upregulation in data from Tusi et al. is suggestive, but not overwhelming. While this could be due to technical difficulties in the single-cell RNA-seq analysis, it might be valuable to calculate the statistical significance of the differences in Fig. 7B between Spi1 levels at the peaks of Cebpa and Gfi1 expression (potential 13.7-9.7) and after those factors are downregulated (potential >100). Also, with regard to the first point, is there any comparison that can be made between these levels of Spi1 and the levels in the Monocyte/Macrophage branches of the SPRING plot?

4. Minor point: in Fig. 7, please change the color of the “zero detected” dots, or else add a note that a black dot in Fig. 7A and D represents failure of detection, not maximal expression.

5. Minor point: The term “data-driven” is applied too narrowly. There is an implication that only whole-genome, bulk population observational studies are a valid source for “data”. But gene network models can establish quite a few biologically important connections using data from other kinds of experiments, especially experiments involving deliberate, focused perturbations that are possible in model systems.

Reviewer #2: The revised manuscript makes some changes in response to some but not all of the concerns from the initial review. After careful re-consideration of the revised manuscript, I agree that the new approach to generate gene circuits from high resolution time course data on gene expression during differentiation of FDCP-mix cells does produce information and some new insights into the magnitude and direction of inputs of the 12 proteins on the 12 genes during that process. However, the conclusions and claims of novel insights into cell fate decisions are over-stated and confusing if not incorrect. One of the problems is the terminology used; in particular it appears that the term differentiation is used for a process more commonly referred to as maturation, i.e. acquisition of the characteristics of mature, circulating blood cells after lineage choice (cell fate) decisions have been executed. Another major issue with the biological interpretation is some aspects of the input data are idiosyncratic, i.e. some of the trends seen in the May et al. data on FDCP-mix cells are not observed during differentiation and maturation of normal blood cells.

(1) The revised version does clarify the differences between the gene circuit models and previous ones. However, the manuscript was not revised the to answer "Exactly what is the output of the gene circuit analysis?" It turns out that the output is the table of interaction coefficients and other parameters. However, those coefficients and parameters are not explained until section 4 (Materials and Methods). It would be more effective and clear for readers to have equation (1) with the explanation of the terms introduced early in the Results.

(2) The manuscript should focus the statements about accuracy and predictions primarily to the FDCP-mix differentiation experiment, and be cautious about conclusions with regard to neutrophil differentiation. The revised manuscript does point out that levels of Gata2 gene expression are much higher in the FDCP-mix experiment than is seen during normal neutrophil differentiation. However, despite the utility FDCP-mix cells as models for several studies, they are not the same as primary, multi-lineage progenitor cells.

(3) The revision did not clarify what values are being graphed in Figure 1 and other figures, but given the values appear to range from 0 to 1.0, and the axis is labeled "Relative expression", perhaps these measurements are fractions of the maximum level observed over the time course. If so, it would help readers to state that explicitly. Perhaps that is what "normalized" in the legend refers to, but normalization can done in many different ways. Knowing clearly what the units are is important, as discussed in the next point.

(4) The manuscript emphasizes that the gene circuit models argue against SPI1 (PU.1) being involved in initiation of the neutrophil cell fate decision, and ascribe that role to CEBPA and GFI1. This conclusion is largely derived from an increase in level of Spi1 mRNA in the later stages of neutrophil "differentiation", while the levels of this mRNA earlier in the time course are considered to be insufficient for SPI1 to play a role in determining the neutrophil cell fate. However, if the scales for mRNA levels refer to "fraction of maximum", then the levels of Spi1 mRNA at early time points appear to be about 25-30% of the maximum, which would be a rather high level. Data from single cell transcriptomes (Fig. S4 and Fig. 7) also show a substantial level of Spi1 mRNA in progenitors, which is also clear in many other transcriptome datasets on hematopoiesis (e.g. from BloodSpot database and servers). These levels of SPI1 are sufficient for this protein to play a role in lineage choice. Examination of Fig. 7 reveals a pattern that is consistent with SPI1 playing an early role in lineage choice. Fig. 7A shows cells with a high probability of differentiating to neutrophils as green, which covers the entire right "leg" of the display. Those cells all have notable expression of the Spi1 gene (as well as Cebpa and Gfi1 genes) denoted as yellow. Thus, the Spi1 gene is expressed at a stage at which it could play a role in lineage choice. The revised manuscript does include a statement on page 23 acknowledging the mutual antagonism between SPI1 and GATA1 in cell fate decision, but the next paragraph argues that the Spi1 gene does not have an initiator role in differentiation. These assertions seem to be counter to each other.

(5) The manuscript should clearly distinguish between lineage-choice (cell fate decisions), which is often referred to as cell differentiation, and cell maturation, which is the acquisition of the characteristics of mature, circulating blood cells after lineage choice. The increase in levels of Spi1 mRNA that is the focus of much of the presentation occurs late in the time-course, long after lineage choice. The manuscript could describe the inferred role of SPI1 as a "hub" (terminology from page 23) to ensure expression of neutrophil-specific genes during the later time points (which themselves are proxies for the later stages of maturation). However, the claims that SPI1 is "not the causative agent of neutrophil development" (p. 3), "Cebpa and Gfi1 are upstream of Spi1 in the genetic architecture of neutrophil differentiation" (page 18), "Spi1 is downstream of Cebpa and Gfi1 in the chain of causation leading to the specification of neutrophils" (page 20), "insight that Cebpa and Gfi1 and not Spi1 are the causal drivers of neutrophil differentiation" (p. 22), and elsewhere are not correct. Those claims could be couched in terms of CEBPA being implicated in the further up-regulation of Spi1 gene expression late in the time course of FDCP-mix cell differentiation, which is a proxy for later stages of neutrophil maturation. Importantly, the data and modeling do NOT argue against a role for SPI1 in neutrophil specification.

(6) As mentioned above, the increase in expression of Gata2 in neutrophils observed in the FDCP-mix transcriptome data of May et al. is not observed in primary cells, as now stated in the revised manuscript (p. 11). The revised manuscript further states that this difference may account for the discrepancy between prediction and observation of the effects of Gata2 knockdown in FDCP-mix cells (Fig. 3 and text on p. 11). However, the model in the manuscript was trained on data from FDCP-mix cells, so the predictions should still be accurate with respect to FDCP-mix, but not with respect to primary cells. The lack of correspondence between the Gata2 RNA data in FDCP-mix cells and primary cells is important to present and discuss, but it would not appear to account for the discrepancy between simulation and observation in FDCP-mix cells. If indeed there is a strong logic for the ability to account for the discrepancy, the manuscript should elaborate on it.

(7) p. 14: After the manuscript presents cases of agreement and conflict between the predictions and experimental result, an overly positive conclusion is drawn: "These inconsistencies were, however, a small proportion of the total inferences and the overall good agreement between model inference and empirical evidence (Fig. 4 and Table S1) suggests a successful decoding of the genetic architecture." A more accurate assessment would be that "...the overall good agreement between model inference and empirical evidence (Fig. 4 and Table S1) suggests a *partially* successful decoding of the genetic architecture."

(8) p. 18: The manuscript concludes that the "activation of neutrophil targets by Gfi1 inferred by the model could, in fact, represent the activity of C/EBPalpha", but elsewhere, including the Abstract and Discussion, that activation is stated as resulting from the input of both proteins. That inconsistency should be corrected.

(9) Top of page 26: The summary conclusion at the end of the Discussion paragraph on inaccuracies in inferences about GATA2 should state clearly that those inferences are incorrect with respect to differentiation in primary cells, not just that users should be "careful about" those inferences.

(10) The labels for panels B, C, and D in Fig. 7 are in a different order from the description in the legend, but it is the same as in the text.

(11) The text is overly long, and it needs to be streamlined.

**Have the authors made all data and (if applicable) computational code underlying the findings in their manuscript fully available?**

Reviewer #1: Yes

Reviewer #2: Yes

PLOS authors have the option to publish the peer review history of their article (what does this mean?). If published, this will include your full peer review and any attached files.

Reviewer #1: No

Reviewer #2: No
---

## [Decision Letter · Decision Letter 2]

21 Dec 2021

Dear Dr. Manu,

We are pleased to inform you that your manuscript 'Data-driven modeling predicts gene regulatory network dynamics during the differentiation of multipotential hematopoietic progenitors' has been provisionally accepted for publication in PLOS Computational Biology.

Best regards,

Stuart A Newman

Guest Editor

PLOS Computational Biology

Ilya Ioshikhes

Deputy Editor

PLOS Computational Biology

Reviewer's Responses to Questions

**Comments to the Authors:**

Reviewer #2: The second revision of this manuscript now draws more conservative conclusions and has been modified to address the several other issues raised in the review of the first revision.

While changes to state more conservative conclusions were made throughout, the Author Summary (page 3) retains the claim about PU.1 not being the "causative agent of neutrophil development":

"Furthermore, our analysis suggested that PU.1, a protein known to be necessary for all white-blood cell lineages, is not the causative agent of neutrophil development, which is driven instead by two other proteins, Cebpa and Gfi1."

The responses from the authors had some misinterpretations of the reviewer comments. Those misinterpretations did not impact the revisions in the manuscript.

For future consideration, the statement (again from the responses, not in the manuscript) that "a necessary condition for a causal or initiating role is that their expression change during differentiation" may not be true for all differentiation decisions. While changes in levels of transcripts and proteins for transcription factors do occur in the bulk of regulatory mechanisms now characterized, regulatory function can change without a change in transcript level, e.g. by post-translational modifications or switches in protein partners.

**Have the authors made all data and (if applicable) computational code underlying the findings in their manuscript fully available?**

Reviewer #2: Yes

PLOS authors have the option to publish the peer review history of their article (what does this mean?). If published, this will include your full peer review and any attached files.

Reviewer #2: **Yes: **Ross C. Hardison

---

## [Editor Report · Acceptance letter]

11 Jan 2022

PCOMPBIOL-D-21-01078R2 

Data-driven modeling predicts gene regulatory network dynamics during the differentiation of multipotential hematopoietic progenitors

Dear Dr Manu,

I am pleased to inform you that your manuscript has been formally accepted for publication in PLOS Computational Biology. Your manuscript is now with our production department and you will be notified of the publication date in due course.

With kind regards,

Livia Horvath
